# Defensin-Rich Platelets Drive Pro-Tumorigenic Programs in Pancreatic Adenocarcinoma

**DOI:** 10.3390/ijms262210898

**Published:** 2025-11-10

**Authors:** Jonathan Gonzalez-Ruiz, Miryam Sarmiento-Casas, Ivan Bahena-Ocampo, Magali Espinosa, Gisela Ceballos-Cancino, Karla Vazquez-Santillan, Vilma Maldonado, Jorge Melendez-Zajgla

**Affiliations:** 1Department of Ciencias de la Salud, Universidad Autónoma Metropolitana-Iztapalapa, Mexico City 09310, Mexico; jonzalezruiz@gmail.com (J.G.-R.); ivan.bahena@gmail.com (I.B.-O.); 2Laboratorio de Genómica Funcional del Cáncer, Instituto Nacional de Medicina Genómica, Mexico City 14610, Mexico; miryam.mood@gmail.com (M.S.-C.); mespinosa@inmegen.gob.mx (M.E.); gceballos@inmegen.gob.mx (G.C.-C.); kivazquez@inmegen.gob.mx (K.V.-S.); 3Laboratorio de Epigenética, Instituto Nacional de Medicina Genómica, Mexico City 14610, Mexico; vmaldonado@inmegen.gob.mx

**Keywords:** pancreatic cancer, platelets, alpha-defensins, *DEFA1/3*, tumor progression, particles like-platelets, zebrafish xenograft, transcriptomics

## Abstract

Pancreatic Ductal Adenocarcinoma (PDAC) is one of the most aggressive and lethal malignancies, driven by late diagnosis, limited therapeutic options, and high metastatic potential. Beyond their canonical roles in hemostasis, platelets have emerged as active modulators of tumor progression and promising noninvasive biomarkers. Among platelet-associated molecules, α-defensins, particularly Defensin Alpha 1/3 (*DEFA1/3*), have been implicated in inflammation and immunity; however, their contribution to PDAC pathogenesis remains unclear. We combined bioinformatic analysis of platelet transcriptomes with functional and in vivo zebrafish xenograft validation to investigate the impact of *DEFA1/3* on PDAC aggressiveness. *DEFA1/3* was significantly upregulated in PDAC-derived platelets. Defensin-enriched platelet-like particles (defensin-rich platelets, DRPs) and recombinant DEFA1/3 enhanced pancreatic cancer cell proliferation, migration, and three-dimensional growth in vitro and promoted tumor dissemination in zebrafish xenografts. Transcriptomic profiling revealed the upregulation of *SPARC*, *KDM6A*, and *GATA6*, whereas clinical data from The Cancer Genome Atlas (TCGA)-PDAC linked high *DEFA1/3* expression to poor survival, increased immune infiltration, and activation of epithelial–mesenchymal transition (EMT). Platelet-derived *DEFA1/3* acts as a functional modulator of PDAC progression, linking platelet granule content to tumor aggressiveness and highlighting a potential biomarker and therapeutic target within the platelet–tumor axis.

## 1. Introduction

Pancreatic Ductal Adenocarcinoma (PDAC) is one of the most aggressive malignancies worldwide [1,2]. In 2022, it accounted for an estimated 510,922 new cases and 467,409 deaths, ranking as the sixth leading cause of cancer-related mortality worldwide [3]. Despite advances in surgery and systemic therapies, the overall five-year survival rate remains close to 10%, reflecting only modest improvement compared to the past decade [4]. Projections indicate that the global incidence will continue to rise, with nearly a 95% increase in new cases by 2050, potentially reaching close to 1 million diagnoses annually [5]. Late diagnosis, intrinsic resistance to chemotherapy, and a highly metastatic phenotype contribute to the poor prognosis of this disease [6].

Beyond tumor-intrinsic drivers, there is increasing evidence highlighting the contribution of systemic factors, such as platelets, to cancer progression [7]. Platelets (PLTs) are small anucleate fragments derived from megakaryocytes (Mks) that play canonical roles in hemostasis and thrombosis [8]. However, in cancer, they acquire additional functions; elevated platelet counts are common in patients with PDAC and are associated with poor prognosis, reduced treatment efficacy, and enhanced metastatic potential [9,10]. The concept of tumor-educated platelets (TEPs) has emerged to describe platelets whose RNA and protein cargo are altered by tumor-derived signals. TEPs not only reflect the systemic influence of the tumor but also actively participate in shaping the tumor microenvironment through secreted mediators and intercellular communication [11,12]. For instance, platelet-derived *ITGA2B* RNA has been identified as a robust biomarker for cancer detection, whereas platelet-secreted Transforming growth factor-beta (TGF-β) acts as a potent inducer of epithelial–mesenchymal transition (EMT), thereby directly promoting tumor progression [12,13].

Defensins are cationic antimicrobial peptides that are part of the innate immune defense [14]. Defensins are classified into three major subfamilies: α, β, and θ. Among them, α-defensins, also known as human neutrophil peptides (HNPs), have been implicated in diverse functions beyond antimicrobial activity, including inflammation and cancer [15]. While β-defensins have been extensively studied and θ-defensins are restricted to certain primates, platelet α-defensins remain comparatively understudied. This limited attention stems from both the technical challenges in their detection and their complex dual role: they can exert antimicrobial and immunomodulatory functions that may either suppress or enhance tumor progression, depending on the biological context [16,17]. In contrast to well-characterized platelet factors such as TGF-β, Platelet factor 4 (PF4), and Vascular Endothelial Growth Factor (VEGF), the biological role of α-defensins in platelets remains largely unexplored. Importantly, platelets and mks have been shown to express Defensin Alpha 1/3 (*DEFA1/3*) transcripts and proteins, which localize to α-granules and are released upon activation [18], suggesting a previously underappreciated immunomodulatory role of platelet-derived α-defensins in the tumor milieu. Thus, it is conceivable that TEPs may carry defensins as part of their altered cargo, potentially contributing to tumor–platelet crosstalk and metastatic conditioning.

Notably, Best et al. demonstrated that the RNA profiles of TEPs can accurately discriminate cancer patients from healthy individuals, including those with pancreatic cancer [19]. However, whether platelet-derived defensins, particularly *DEFA1/3*, contribute to PDAC progression has not been addressed. To explore this question in a physiologically relevant context, we employed zebrafish (*Danio rerio*) xenografts, a well-established model that enables real-time visualization of human cancer cell migration and reproduces conserved platelet–tumor interactions observed in humans. This study aimed to determine whether platelet-derived *DEFA1/3* contributes to the aggressiveness of PDAC. Specifically, we hypothesized that platelets enriched in *DEFA1/3* promote pro-tumorigenic and pro-migratory programs in PDAC cells. To test this hypothesis, we combined platelet transcriptomic re-analysis, in vitro functional assays, and in vivo zebrafish xenografts. Despite increasing evidence linking platelet-derived factors to tumor progression, the role of platelet defensins in pancreatic cancer biology remains largely unexplored. Here, we report that defensin-rich platelets (DRPs) promote proliferation, migration, and spheroid growth of pancreatic cancer cells, upregulate key aggressiveness-related genes, and are associated with adverse clinical outcomes, supporting the concept that platelet-derived *DEFA1/3* represents a functional driver for PDAC progression and a candidate for translational exploration as a biomarker or therapeutic target.

## 2. Results

### 2.1. Transcriptomic Alterations in PDAC-Derived Platelets

For this study, we re-analyzed the platelet RNA-sequencing dataset originally reported by Best et al., in which circulating platelets were profiled from patients with cancer and healthy donors and subsequently used to train a machine learning algorithm for cancer detection. Rather than focusing on prediction, we curated a subset of this dataset to investigate transcriptional alterations in PDAC. Our analysis included platelets isolated from patients with PDAC (*n* = 35) and healthy controls (*n* = 55). Raw RNA-seq counts from platelet samples were retrieved and processed using the Correlative Analyses of Gene Set Enrichment (CAGE) framework. Counts were normalized to counts per million (CPM) and transformed using voom, enabling variance-stabilized differential expression analysis. This approach revealed a distinct distribution of differentially expressed genes between PDAC-derived platelets and those from healthy donors (Figure 1a), indicating the transcriptional reprogramming of circulating platelets. The heatmap revealed a distinct cluster of genes significantly upregulated in PDAC-derived platelets compared to those from healthy donors, revealing a cancer-specific transcriptional signature. To contextualize these findings, differentially expressed genes were compared with a curated panel of established cancer biomarkers, which revealed a significant overlap of 94 genes (18.8%). Among these, several members of the defensin family, including *DEFA8P* (fold change = 16.87), *DEFA4* (fold change = 11.37), and *DEFA1B* (fold change = 10.14), were consistently overexpressed (Figure 1b). Defensins are well-known mediators of innate immunity and epithelial defense, suggesting a potential role for these molecules in platelet-associated immune modulation within the tumor microenvironment. Gene Ontology (GO) enrichment analysis further highlighted the biological context of these transcriptional changes. The most significantly enriched pathway was regulated exocytosis (GO:0045055), followed by processes related to cellular homeostasis, cytokine production, lymphocyte migration, and negative regulation of leukocyte differentiation. These results underscore the broad alterations in the immune and secretory functions of PDAC-derived platelets. Notably, enrichment was also observed in megakaryocyte development and platelet production pathways, indicating a potential link between tumor progression and hematopoietic modulation (Figure 1c). Together, these findings provide an initial indication that platelet defensins, particularly *DEFA* family members, represent active components of platelet–tumor communication in PDAC, forming the basis for our subsequent functional validation.

### 2.2. Generation of an In Vitro Model of DRPs

Building on these transcriptomic findings, we sought to determine whether platelet defensins exert functional effects on PDAC biology using a controlled in vitro model. To validate the transcriptomic findings and establish an experimental platform for functional studies, we generated an in vitro model of platelet production (Figure 2a). MEG-01 megakaryoblastic cells were differentiated into Platelet-Like Particles (PLPs) following recombinant human thrombopoietin (TPO; Abcam, Cambridge, UK) stimulation, thereby providing a renewable source of platelet surrogates (Appendix A). To specifically assess the role of *DEFA1/3* in platelet biology, MEG-01 cells were genetically modified via lentiviral transduction to stably overexpress *DEFA1/3* (Figure 2b). These cells were subsequently differentiated, yielding two distinct preparations: control Platelet-Like Particles Empty vector (PLP-Ev) and DRPs.

Functional assays demonstrated that PLPs generated in vitro retained biological activity comparable to that of peripheral blood-derived platelets. When co-cultured with BxPC-3 pancreatic cancer cells, PLPs significantly increased cell viability relative to non-treated controls, mirroring the effect observed with donor-derived platelets (Figure 2c). Notably, DRPs had an even greater effect on tumor cell viability, suggesting that *DEFA1/3* enrichment enhances the pro-tumorigenic potential of platelet-derived particles.

Molecular validation confirmed that the DRPs contained significantly elevated levels of *DEFA1/3.* Quantitative PCR (qPCR) analysis revealed robust increases in *DEFA1/3* mRNA compared to PLP-Ev and peripheral blood platelets, while Western blot analysis of in vitro–differentiated PLPs and DRPs corroborated these findings at the protein level, confirming higher DEFA1/3 content in DRPs (Figure 2d,e). Collectively, these results establish a reliable in vitro model of platelet production. As an initial validation, we tested the direct effect of recombinant human DEFA1/3 on the viability of several pancreatic cancer cell lines. Increasing concentrations of DEFA1/3 protein resulted in a dose-dependent increase in cell viability in BxPC-3, AsPC-1, and Capan-2 cells, with significant effects observed at concentrations ≥ 100 ng/mL (Appendix A). These results support the hypothesis that DEFA1/3 enhances tumor cell growth and provide a rationale for subsequent experiments using DRPs. These results validated the transcriptomic observations and established a mechanistic framework to test whether DRPs modulate cancer cell migration and aggressiveness.

### 2.3. DRPs Enhance the Migratory Capacity of Pancreatic Cancer Cells

Given the established link between platelet activation and tumor dissemination, we assessed whether DRPs influence the migratory behavior of PDAC cells. Next, we investigated the impact of DRPs on the migratory capacity of pancreatic cancer cells. BxPC-3 and AsPC-1 cells were cultured in Ibidi^®^ chamber inserts to generate standardized wounds and were subsequently treated with control PLP-Ev or DRPs. To exclude the contribution of cell proliferation, all assays were performed in the presence of the Deoxyribonucleic Acid (DNA) synthesis inhibitor Ara C (15 μM). Representative phase-contrast images captured at baseline (T0) and after 72 h revealed accelerated wound closure in DRP-treated cultures compared to in the non-treated and PLP-Ev groups (Figure 3a,b). Quantification of the uncovered wound area confirmed that DRP treatment significantly enhanced the migratory capacity of both AsPC-1 and BxPC-3 cells (Figure 3c,d). Notably, BxPC-3 cells treated with DRPs exhibited near-complete wound closure within 72 h, underscoring the potent pro-migratory effect of the DRPs particles. Collectively, these findings indicate that DRPs provide a strong stimulus for pancreatic cancer cell migration, independent of cell proliferation.

### 2.4. DRPs Promote Proliferation, 3d Spheroid Growth, and Clonogenic Potential in Pancreatic Cancer Cells

As enhanced migration is often accompanied by increased proliferative and survival capacity, we investigated whether DRPs also promote PDAC cell growth and long-term expansion. These findings strongly support the notion that DRPs exert pro-tumorigenic effects. To further investigate this hypothesis, we performed complementary assays using BxPC-3 pancreatic cancer cells. First, a cell proliferation assay was conducted 72 h after treatment with DRPs or control PLPs. DRPs significantly increased cell proliferation compared to non-treated and PLP-Ev conditions (Figure 4a), indicating that *DEFA1/3* enrichment enhances the proliferative support provided by platelet-like particles.

Next, we evaluated the three-dimensional growth capacity using a spheroid formation assay. BxPC-3 cells were cultured under non-adherent conditions and treated with recombinant DEFA1/3 protein, DRPs, or control agents for 10 days. By day 5, both DEFA1/3- and DRP-treated cultures exhibited a marked increase in spheroid number compared to untreated cells, with DRPs showing the strongest effect (~50% increase) (Figure 4b,c). In addition, spheroids formed under either DEFA1/3 or DRP treatment displayed a significant increase in size, with mean diameters approximately 30% larger than those observed in the control groups (Figure 4d). Together, these results demonstrate that DRPs promote both proliferation and sustained three-dimensional tumor growth, further underscoring their pro-tumorigenic potential. To further evaluate the long-term proliferative potential conferred by DRPs, clonogenic assays were performed in BxPC-3, AsPC-1, and Capan-2 cells using crystal violet staining. DRP treatment significantly increased the number and density of colonies in all three cell lines, although the effect was less pronounced in Capan-2 cells (Figure 5a–c). Altogether, these results demonstrate that DRPs not only enhance short-term proliferation but also sustain three-dimensional growth and clonogenic capacity across multiple PDAC cell lines, reinforcing their role as potent pro-tumorigenic modulators of PDAC cells.

To assess whether these effects were mediated by direct interactions with platelet-like particles or soluble factors released into the culture medium, cells were exposed to conditioned media obtained from PLP-Ev and DRP cultures. Interestingly, the DRP-conditioned medium induced a moderate but consistent increase in colony formation across all three cell lines, albeit to a lesser extent than that observed with direct DRP treatment. These findings suggest that *DEFA1/3* is not only retained within platelet-like structures but also released into the extracellular environment, where it contributes to the pro-tumorigenic activity of DRPs.

### 2.5. DRPs Enhance Tumor Growth and Dissemination In Vivo

To verify whether the pro-tumorigenic effects of DRPs observed in vitro translate to a living system, we extended our analysis using a zebrafish xenograft model of human cancer. To extend these in vitro findings to an in vivo context, we employed a zebrafish xenograft model. BxPC-3 pancreatic cancer cells pretreated with PLP-Ev or DRPs were injected into the yolk sac of zebrafish embryos at 2 days post-fertilization. Following incubation, control PLP-Ev-treated cells formed larger tumor masses than the untreated controls and exhibited evidence of migration beyond the injection site. Notably, DRP-treated cells generated larger tumor volumes and displayed significantly enhanced migratory behavior, as evidenced by the dissemination of fluorescent cells to distal regions, including the head and tail (Figure 6a). Although representative images may not fully capture the extent of cellular density within the yolk sac owing to technical variability in fluorescence acquisition, quantitative analysis confirmed that the DRP-treated xenografts exhibited the greatest tumor volumes and dissemination levels.

Consistently, quantification of tumor burden and migratory spread confirmed that DRP treatment significantly increased both tumor volume and the number of disseminated cells relative to the controls (Figure 6b,c). These results validate the in vitro data and demonstrate that DRPs not only promote pancreatic cancer cell growth but also enhance their invasive potential in vivo. Collectively, these functional studies consistently demonstrate that DRPs confer a strong pro-tumorigenic advantage to pancreatic cancer cells. Across multiple experimental platforms, including proliferation and clonogenicity assays, wound healing migration assays, three-dimensional spheroid cultures, and zebrafish xenografts, DRP treatment consistently enhanced tumor cell viability, migratory capacity, and invasive growth. Importantly, these effects were more pronounced than those observed with control PLPs or platelet-conditioned medium, indicating the effect of *DEFA1/3* enrichment on platelet-derived particles.

Collectively, these findings highlight DRPs as potent modulators of pancreatic cancer progression and provide a robust functional framework for subsequent molecular and clinical analyses of DRPs. These in vivo data support our mechanistic findings, demonstrating that *DEFA1/3*-enriched platelets potentiate PDAC aggressiveness across biological scales.

### 2.6. DRPs Induce Transcriptional Reprogramming in PDAC Cells

Having established that *DEFA1/3*-enriched platelets promote PDAC growth and dissemination in vivo, we next examined whether these effects were mirrored at the molecular level and reflected in patient-derived transcriptomic data from the TCGA database. To mechanistically link our experimental observations with clinical and transcriptomic evidence, we performed an integrated in vitro and in silico analysis of the data. (Figure 7a–d) qPCR analysis showed that DRP-treated BxPC-3 cells exhibited a significant upregulation of *SPARC*, *KDM6A*, and *GATA6* compared to the Nt group (*p* < 0.001), while no significant changes were observed in *EGR1* expression. These results suggest that DRPs selectively activate transcriptional programs associated with the migratory and invasive potential of PDAC.

To assess the clinical relevance of these observations, we analyzed OS in the TCGA-PDAC cohort stratified by high/low *DEFA1/3* expression and platelet counts. Patients with low *DEFA1/3* and high platelet levels exhibited the most favorable outcomes, whereas those with high DEFA1/3 and low platelet levels exhibited the poorest survival (log-rank *p* = 0.016) (Figure 7e). ssGSEA analysis further demonstrated that *DEFA1/3* expression positively correlated with platelet signatures (rho = 0.16, *p* = 0.03), neutrophil infiltration (rho = 0.36, *p* = 1 × 10^−6^), and EMT activation (rho = 0.20, *p* = 0.009) (Figure 7f–h). These associations provide a mechanistic link between *DEFA1/3* expression, immune modulation, and enhanced tumorigenic plasticity.

Collectively, these findings demonstrate that DRPs induce the expression of genes linked to aggressive PDAC phenotypes and that DRPs expression is associated with poor prognosis, immune cell infiltration, and EMT activation in patient datasets.

### 2.7. Correlation of SPARC, GATA6, and KDM6A with Aggressive Transcriptional Programs in TCGA-PDAC

To further explore the molecular programs associated with DRP-regulated genes, we analyzed the correlations between *SPARC*, *GATA6*, and *KDM6A* expression and transcriptional signatures related to invasion, motility, and EMT in TCGA-PDAC samples. (Appendix A) Invasion-related MMP score showed a strong positive correlation with *SPARC* (rho = 0.821, *p* = 1.9 × 10^−44^), whereas only weak or non-significant associations were observed with *GATA6* (rho = −0.212, *p* = 0.0471) and *KDM6A* (rho = −0.083, *p* = 0.276) (Appendix A). Motility-integrin score was significantly negatively correlated with *GATA6* (rho = −0.306, *p* = 2.81 × 10^−5^) and strongly positively correlated with *SPARC* (rho = 0.565, *p* = 2.46 × 10^−16^), while no significant correlation was found for *KDM6A* (rho = −0.0847, *p* = 0.262). (Appendix A) EMT-like score also showed a robust positive correlation with *SPARC* (rho = 0.803, *p* = 4.76 × 10^−85^) and a significant negative correlation with *GATA6* (rho = −0.336, *p* = 4.86 × 10^−6^), whereas *KDM6A* expression did not correlate with EMT signatures (rho = 0.0042, *p* = 0.956).

Overall, these analyses revealed that *SPARC* expression was strongly associated with invasive, motile, and EMT-like transcriptional programs, consistent with its role as a mediator of tumor progression, whereas *GATA6* showed an inverse relationship with motility and EMT, in line with its reported function in less aggressive PDAC subtypes. Together with our functional results, these correlations suggest that DRPs rewire transcriptional programs associated with invasion and plasticity, supporting the central hypothesis that platelet-derived *DEFA1/3* promotes pro-tumorigenic reprogramming in PDAC.

## 3. Discussion

Defensins are a diverse family of cationic antimicrobial peptides whose functions extend beyond host defense into cancer biology, where they display context-dependent pro- or antitumorigenic effects [20]. Although β-defensins have been studied more extensively, the roles of α-defensins, particularly in PDAC, remain comparatively underexplored [21,22,23]. In this study, we identified platelet-derived α-defensins (DEFA1/3) as active modulators of PDAC aggressiveness. We propose that these cationic peptides, delivered via platelet granules, reshape the transcriptional landscape of cancer cells to promote malignant phenotypes, thereby providing a mechanistic link between platelet biology and tumor progression [14,15,24]. Although our analyses revealed strong associations between platelet-derived defensins and PDAC aggressiveness, these correlations alone do not establish causality. Therefore, we combined bioinformatic inference with functional experiments to directly assess the biological impact of these mutations.

Re-analysis of platelet RNA-seq data from PDAC patients by Best et al. [19] revealed defensins among the most upregulated transcripts, alongside the enrichment of regulated exocytosis pathways. To functionally validate these observations, we generated *DEFA1/3*-rich platelet-like particles using MEG-01 cells. Exposure to DRPs increased proliferation, clonogenicity, migration, and three-dimensional spheroid growth in multiple PDAC cell lines. Importantly, migration assays performed under DNA replication blockade confirmed that wound closure reflected enhanced motility rather than proliferative expansion of cells. These results align with established literature demonstrating that platelets facilitate tumor cell migration and invasion through mechanisms such as TGF-β secretion, induction of EMT, and P-selectin-mediated adhesion [1,19,25]. Additional studies have shown that platelet Panx1 drives PDAC cell invasion and that the release of nucleotides adenosine triphosphate (ATP) and adenosine diphosphate (ADP) facilitates transendothelial migration and metastatic spread [26,27,28]. Our data extend these paradigms by implicating platelet-derived defensins as potent effectors of platelet–tumor cross-talk.

In vivo validation using zebrafish xenografts further demonstrated that DRP treatment increased tumor burden and dissemination, confirming that the pro-tumorigenic activity of platelet-derived defensins extends into a complex microenvironment [29,30]. Zebrafish models are particularly suited for real-time visualization of tumor–platelet–microenvironment interactions, such as invasion, vascular remodeling, and immune escape [29,31,32]. Consistent with prior reports, platelet depletion in murine models reduces the metastatic burden [33,34,35], highlighting the causal role of platelets in dissemination. Together, these results support the view that *DEFA1/3*-rich platelets act not only as passive biomarkers but also as active facilitators of PDAC progression.

At the molecular level, DRP exposure selectively upregulated *SPARC*, *KDM6A*, and *GATA6* transcripts in PDAC cells, whereas EGR1 levels remained unchanged. Analysis of TCGA-PDAC confirmed clinical relevance, showing that high *DEFA1/3* expression combined with elevated platelet counts correlated with poorer overall survival and was associated with platelet signatures, neutrophil infiltration, and EMT activation. These transcriptomic and survival associations are inherently correlative; however, their consistency with our mechanistic data reinforces the notion that platelet-derived defensins contribute to PDAC progression. Comparisons with other cancers underscore the specificity of these findings to PDAC. *SPARC* has been linked to stromal remodeling and poor prognosis in hepatocellular carcinoma and breast cancer, but its role in desmoplasia is particularly pronounced in PDAC [26,36,37,38]. *KDM6A*, frequently mutated in bladder and lung cancers [39,40], has been associated with lineage flexibility and aggressive basal-like phenotypes in PDAC [41,42]. *GATA6*, known to regulate differentiation in gastric and colorectal cancers, plays a defining role in PDAC subtype stratification with major prognostic implications [43,44]. In contrast, *EGR1*, an oncogenic driver in prostate and lung cancers, did not change [45,46] in our system, indicating a more PDAC-specific transcriptional program activated by platelet-derived defensins.

### Strengths and Limitations

This study has several strengths: (i) a rigorous bioinformatic re-analysis that anchors the work in publicly available evidence; (ii) a renewable MEG-01–derived platelet platform that yields standardized DRPs and enables reproducibility; (iii) orthogonal functional assays spanning 2D and 3D systems; (iv) in vivo validation to confirm biological relevance; and (v) correlative analyses across patient datasets that connect the mechanism to the clinical context. This multi-layered, cross-validated design substantially enhanced the robustness, internal consistency, and translational relevance of our conclusions.

The limitations are modest and primarily relate to scope: confirmation with patient-derived platelets would further cement physiological relevance; direct causality for *DEFA1/3* is best addressed with genetic or neutralization strategies in immunocompetent mammalian models; and bulk RNA-seq can be influenced by stromal content and systemic inflammation. Although the zebrafish xenograft model is advantageous for real-time visualization of tumor cell dissemination and drug-free microenvironmental interactions, it does not fully recapitulate the complexity of the human immune system or the stromal context of pancreatic tumors. Moreover, quantification of fluorescent cell migration, although standardized and performed under blinded conditions, remains semi-quantitative and restricted to short-term observations (24 h post-injection). Finally, the relatively small number of biological replicates inherent to in vivo zebrafish assays may introduce experimental variability. Despite these constraints, the concordance between our in vitro and in vivo findings strengthens the interpretation that Defensin-Rich Platelets actively promote pro-tumorigenic phenotypes in PDAC. These points outline clear next steps and do not detract from the core findings of this study.

Taken together, our data position platelet-derived *DEFA1/3* as active contributors to the malignant PDAC phenotype. By linking platelet granule cargo with epithelial plasticity, immune infiltration, and stromal remodeling, we provide evidence that tumor-educated platelets function as mediators, not mere bystanders, of tumor progression, broadening the current understanding of the platelet–tumor axis and highlighting actionable avenues for mechanistic and translational follow-up. By integrating correlative and functional evidence, our study delineates the role of platelet defensins as both biomarkers and active modulators of tumor behavior.

## 4. Materials and Methods

### 4.1. Cell Culture

The human 293T, AsPC-1, BxPC-3, Capan-2, and MEG-01 cell lines were obtained from the American Type Culture Collection (ATCC, Manassas, VA, USA). All cell lines were authenticated by the ATCC. Cells were cultured according to the ATCC recommendations. 293T cells were maintained in Dulbecco’s Modified Eagle Medium (DMEM; ATCC 30-2002) supplemented with 10% (*v*/*v*) FBS (Gibco, Cat. No. 16000044; Thermo Fisher Scientific, Waltham, MA, USA) and 1% penicillin–streptomycin (10,000 U/mL; Cat. No. 15140122. Thermo Fisher Scientific, Waltham, MA, USA). AsPC-1, BxPC-3, and MEG-01 cells were cultured in RPMI-1640 medium (ATCC 30-2001) containing 10% FBS and 1% penicillin–streptomycin. Capan-2 cells were cultured in McCoy’s 5A medium (ATCC 30-2007) supplemented with 10% FBS and 1% penicillin–streptomycin. All cultures were maintained at 37 °C in a humidified incubator with 5% CO_2_ and passaged every 2–3 days to maintain exponential growth. For all experiments, cells were used between passages 5 and 20 to ensure consistency. Before treatment, cells were plated in T-75 flasks (Corning, Corning, NY, USA) or 6-well plates (Sigma-Aldrich, St. Louis, MO, USA) and allowed to adhere overnight. The following day, the culture medium was replaced with a serum-free medium for 12 h to synchronize growth and minimize serum-derived variability. For defensin stimulation assays, cells were treated with recombinant human Defensin Alpha 1/3 (DEFA1/3; Abcam, Cambridge, UK, Cat. #ab97409) at a final concentration of 100 ng/mL for 72 h. Control (non-treated; Nt) cultures received the vehicle alone under identical conditions. All treatments were performed in biological triplicates, and each experiment was independently repeated at least three times to ensure reproducibility. All cell lines were mycoplasma-free.

### 4.2. Sample Classification

To classify the platelet transcriptomic profiles of patients and controls, we implemented a Support Vector Machine (SVM) algorithm using the e1071 R package (v1.7-13) within R version 4.3.0 (R Foundation for Statistical Computing, Vienna, Austria). Normalized RNA-seq expression matrices were imported into R, and low-abundance genes (CPM < 1 in >80% of samples) were filtered out to reduce noise. The remaining transcripts were log_2_-transformed and scaled to zero mean and unit variance before model training. In the classification procedure, the SVM algorithm projected each sample into a high-dimensional feature space, where each axis represented a transcript. The position of a given sample along each axis corresponds to the normalized expression value of that gene. A One-Versus-One (OVO) strategy was applied to handle multiclass classification. In this approach, a separate binary SVM model is trained for each pair of classes, and the final class assignment for a given sample is determined by majority voting across all pairwise classifiers. The kernel function used was the radial basis function (RBF), which efficiently captures the nonlinear relationships between gene expression profiles. The cost (C) and gamma (γ) parameters were optimized using a 10-fold cross-validation within a grid search framework (tune.svm function). To evaluate the model performance, we used accuracy, F1-score, and area under the ROC curve (AUC), which were computed using the caret (v6.0-94) and pROC (v1.18.5) R packages. The misclassification error was assessed using confusion matrices generated from the withheld test samples (20% of the dataset, randomly selected). All analyses were conducted using reproducible R scripts, and the random seed was fixed at set.seed (1234) to ensure consistent classification results.

### 4.3. Differentiation of Mks (MEG-01) Cells into Platelet-like Particles

To obtain PLPs, MEG-01 megakaryoblast cells were differentiated in vitro following an optimized protocol adapted from Risitano et al., 2012 [47] with minor modifications. Briefly, MEG-01 cells were seeded at a density of 2 × 10^5^ cells/mL in RPMI-1640 medium supplemented with 10% FBS and 1% penicillin/streptomycin. Recombinant human thrombopoietin (TPO) (PeproTech No. 300-18; Rocky Hill, NJ, USA) was added to the cultures at a final concentration of 100 ng/mL, and the cells were maintained at 37 °C in a humidified incubator with 5% CO_2_. The differentiation process spanned 7 days, during which fresh TPO (100 ng/mL) was added every third day to sustain the megaryocytic maturation and proplatelet formation. Cell morphology was monitored daily using phase-contrast microscopy to confirm the appearance of cytoplasmic extensions and platelet-like protrusions. Following the differentiation period, the PLPs were purified using a stepwise centrifugation protocol. The cultures were centrifuged at 100× *g* for 5 min to remove intact cells. The supernatant was carefully transferred and centrifuged again at 150× *g* for 15 min to eliminate larger debris and fragments. Finally, PLPs were pelleted by centrifugation of the resulting supernatant at 750× *g* for 15 min [18]. The resulting pellet was resuspended in serum-free RPMI and quantified for protein concentration (Bradford assay) and particle counting using a Neubauer chamber under light microscopy. PLP morphology and purity were further verified by Giemsa staining prior to downstream assays.

### 4.4. Isolation of PTLS from Healthy Donor Samples

PLTs were isolated from voluntary laboratory donors under sterile conditions. Whole blood was collected into BD Vacutainer^®^ sodium citrate tubes (Becton Dickinson, Franklin Lakes, NJ, USA) to prevent coagulation, and samples were processed within 2 h of collection to ensure platelet integrity. Platelet-rich plasma (PRP) was obtained by centrifuging whole blood at 100× *g* for 20 min at room temperature (RT) without a break. The upper PRP layer was carefully transferred to a new tube, and acetylsalicylic acid was added to a final concentration of 200 μM to inhibit the platelet activation. The PRP was centrifuged again at 100× *g* for 20 min to eliminate residual red blood cells. Subsequently, the platelets were pelleted by centrifugation at 800× *g* for 20 min at RT, and the supernatant was discarded. The resulting platelet pellet was gently resuspended in Tyrode’s buffer (pH 7.4) containing 0.1% bovine serum albumin (BSA) (Sigma-Aldrich; Cat. No. A2153). Platelet concentration and purity were assessed by manual counting using a Neubauer hemocytometer under light microscopy, and morphology was verified by phase-contrast imaging [48]. Purified PLTs were used immediately after isolation for all functional assays. PRP samples were obtained from voluntary laboratory donors who provided peripheral blood exclusively for in vitro research. No personal or clinical data were collected, and all samples were anonymized before processing. The use of human blood-derived material was conducted in accordance with institutional biosafety and ethical guidelines, under minimal-risk classification, and did not require formal informed consent according to national and international regulations (NOM-012-SSA3-2012 [49]; CIOMS, 2016 [50]).

### 4.5. Generation of DEFA1/3-Rich Platelet-Derived Particles

To generate DRPs, the megakaryoblastic cell line MEG-01 was genetically engineered using lentiviral transduction to overexpress *DEFA1/3*. Lentiviral particles were produced by co-transfecting 293T cells with Lipofectamine™ 2000 (Thermo Fisher Scientific, Cat. No. 11668019; Waltham, MA, USA) with the following plasmids: pLV-IRES-*DEFA1/3* (donor vector containing the *DEFA1/3* coding sequence inserted into the pLV-IRES backbone), pLJM1-GFP (transduction efficiency control), psPAX2 (Addgene, plasmid #12260; Watertown, MA, USA) (packaging plasmid), and pCMV-VSV-G (Addgene, plasmid #8454; Watertown, MA, USA) (envelope plasmid). Transfected cells were maintained in DMEM with 10% FBS for 48 h, and viral supernatants were collected every 12 h for 3 d, filtered through a 0.45 μm PES filter, and stored at 4 °C until use. MEG-01 cells were seeded at 2 × 10^5^ cells/mL and transduced with the viral supernatants at a multiplicity of infection (MOI) of 100, in the presence of 8 μg/mL polybrene to enhance infection efficiency. After 48 h, the medium was replaced, and the cells were cultured for an additional 72 h before puromycin (1 μg/mL) was applied for selection [51]. Stable populations were obtained after three rounds of selection, yielding approximately 80% transduction efficiency as verified by GFP fluorescence. Following selection, MEG-01-*DEFA1/3* and MEG-01-EV (empty vector control) cells were stimulated with recombinant TPO (200 ng/mL) for 72 h to induce the formation of PLPs. The released particles were collected by differential centrifugation, as described in Section 4.3, and designated as DRPs (*DEFA1/3*-rich platelets) and control platelets (PLPs-EV). Overexpression of *DEFA1/3* in MEG-01-*DEFA1/3* cells and DRPs was confirmed by qPCR and Western blot analysis prior to downstream functional assays.

### 4.6. Genomic Studies

Total RNA was extracted, and RT-PCR analysis was performed as previously described by Bandala et al. (2001) [52], with minor modifications to ensure reproducibility. Briefly, total RNA was extracted from AsPC-1, BxPC-3, MEG-01, PLPs-Ev, and DRPs cells using TRIzol™ reagent (Invitrogen, Waltham, MA, USA. No. 15596026, USA) according to the manufacturer instructions. RNA concentration and purity were determined using a NanoDrop™2000c (Thermo Fisher Scientific, Waltham, MA, USA) spectrophotometer by measuring the A260/A280 ratio, and integrity was verified by 1% agarose gel electrophoresis. For reverse transcription, 1 μg of total RNA was retrotranscribed in a 20 μL final reaction volume using ThermoScript™ Reverse Transcriptase (Invitrogen) with random hexamer primers and reaction buffer supplied by the manufacturer (Invitrogen). Polymerase Chain Reaction (PCR) amplifications were performed using AmpliTaq Gold™ DNA polymerase (Applied Biosystems) in a 25 μL reaction volume containing 2.5 μL of 10× PCR buffer, 0.5 μL of dNTP mix (10 mM each), 1 μL of forward and reverse primers (10 μM each), and 1 μL of complementary DNA (cDNA) template. Thermal cycling was performed in a Veriti™ 96-Well Thermal Cycler (Applied Biosystems, Foster City, CA, USA) under the following conditions: initial denaturation at 95 °C for 10 min, followed by 30 cycles of 95 °C for 30 s, 50 °C for 45 s, and 72 °C for 1 min. Final extension: 72 °C for 5 min. PCR products were electrophoresed on 1–2% agarose gels stained with SYBR™ Safe DNA Gel Stain (Invitrogen, No. S33102) and visualized under UV light. Bands corresponding to the expected amplicon sizes were excised and purified using the QIAquick Gel Extraction Kit (Qiagen GmbH, Hilden, Germany) for Sanger sequencing. Amplification efficiency and linearity were verified by plotting standard curves derived from serial dilutions of cDNA, confirming that the reactions proceeded within the logarithmic phase. The list of primers is provided in Appendix A.

### 4.7. Transfection

For lentiviral particle production, 293T cells were co-transfected with the following plasmids: donor vector pLV-IRES-*DEFA1/3*, control vector pLJM1-*GFP*, packaging plasmid psPAX2, and envelope plasmid pCMV-VSV-G, following the manufacturer’s instructions for Lipofectamine™ 2000 (Invitrogen, USA). Before transfection, plasmid DNA was purified using an endotoxin-free Maxiprep kit (Qiagen, Cat. No. 12362. Germany) to ensure optimal transfection efficiency and to prevent cytotoxicity. For each 10 cm plate, 12 μg of total DNA (ratio 4:3:2:1 for pLV-IRES-*DEFA1/3*: psPAX2: pCMV-VSV-G: pLJM1-*GFP*) was mixed with Lipofectamine reagent in Opti-MEM™ medium and incubated for 20 min at room temperature before being added to the cells. After an 8 h co-transfection period, the transfection medium was replaced with complete DMEM supplemented with 10% FBS, and the cells were incubated at 37 °C and 5% CO_2_. Viral supernatants were collected at 48 and 72 h post-transfection, clarified by centrifugation at 500× *g* for 10 min, and filtered through a 0.45 μm PES filter to remove cellular debris. The viral suspension was then concentrated by ultracentrifugation at 25,000× *g* for 2 h at 4 °C using a SW32 Ti rotor (Beckman Coulter). The resulting viral pellet was resuspended in 1 mL of sterile Phosphate-Buffered Saline (PBS) containing 1% BSA and stored at −80 °C in single-use aliquots to avoid repeated freeze–thaw cycles. Viral titers were determined by transducing 293T cells with serial dilutions of the viral stock and quantifying GFP-positive cells after 72 h using fluorescence microscopy, as described by Schwarz-Cruz et al., 2020 [53]. To determine the optimal transduction conditions, AsPC-1 cells were initially seeded to reach 80% confluence and infected with Lv-GFP lentiviral particles at different multiplicities of infection (MOI = 10, 20, 50, 100, and 200). After 48 h, GFP expression was evaluated using a Leica fluorescence microscope (Wetzlar, Germany). An MOI of 100 was selected as the optimal value for subsequent experiments, as it achieved robust transgene expression while minimizing cytotoxicity. Transduction was performed at 37 °C for 24 h, after which the virus-containing medium was replaced with fresh complete RPMI medium supplemented with 10% FBS. The cells were then incubated for an additional 48 h before initiating antibiotic selection. To generate stable cell lines, puromycin (2 μg/mL; Sigma-Aldrich, St. Louis, MO, USA) was added to the culture medium, which was refreshed every two days. Selection continued until all non-infected control cells were no longer viable (approximately 5–7 days). Surviving clones were expanded and maintained in 1 μg/mL puromycin to ensure stable transgene expression. The resulting cell populations were designated as MEG-01-*DEFA1/3* and MEG-01-*GFP*, corresponding to cells stably expressing the DEFA1/3 transgene or GFP control, respectively [53].

### 4.8. Cell Migration Assays

To evaluate the effects of DEFA1/3 and platelet-derived particles on the migratory capacity of pancreatic cancer cells, AsPC-1 and BxPC-3 cells were used. A total of 4 × 10^5^ cells were seeded into (Ibidi^®^ 35 mm Culture Dishes containing Culture-Insert 2 Well systems (Cat. No. 80206; Ibidi GmbH, Gräfelfing, Germany) and allowed to form confluent monolayers within 24 h. Each insert well received 70 μL of cell suspension and was incubated at 37 °C in a humidified atmosphere containing 5% CO_2_. To ensure that wound closure reflected cell migration rather than proliferation, the medium was supplemented with 5 μM cytosine β-D-arabinofuranoside hydrochloride (Ara-C; Sigma-Aldrich). Cat. No. C1768; St. Louis, MO, USA) as a DNA synthesis inhibitor. After the initial 24 h attachment period, the Culture-Insert 2 Well was gently removed using sterile forceps to create a defined cell-free gap (wound). Cells were then treated under the following experimental conditions: Recombinant DEFA1/3 (100 ng/mL; Abcam plc, Waltham, MA, USA), PLP-Ev, DRPs, and Nt. The cultures were maintained for an additional 72 h under standard conditions. Wound closure was monitored by capturing five random fields per condition using an inverted phase-contrast microscope (Leica Microsystems, Wetzlar, Germany) at 0, 24, 48, and 72 h. Images were analyzed using the ImageJ software version 1.53t (NIH, Bethesda, MD, USA). The percentage of wound closure was calculated as follows:$$\mathrm{Migration}\mathrm{rate}=\left(1-\frac{\mathrm{wound}\mathrm{area}\mathrm{at}\mathrm{time}t}{\mathrm{initial}\mathrm{wound}\mathrm{at}0\;h}\right)\times 100$$

Each condition was tested in triplicate, and the results are presented as mean ± S.E.M from at least three independent experiments.

### 4.9. Spheroid Culture

Capan-2 and BxPC-3 pancreatic cancer cells were cultured using the liquid overlay technique, as previously described by Espinosa et al. (2012) [54], with minor modifications. Briefly, P60 Petri dishes were coated with a thin layer of 1% agarose (Sigma-Aldrich, Cat. No. A0169; St. Louis, MO, USA) (*w*/*v*) prepared in sterile PBS to prevent cell adhesion and allowed it to solidify at room temperature. Subsequently, 1 × 10^6^ cells, previously expanded as a monolayer, were seeded onto each agarose-coated plate in Leibovitz L-15 medium supplemented with 5% FBS (Gibco) and incubated at 37 °C in a humidified atmosphere. After 48 h, the floating cell aggregates began to form compact spheroids. Cultures were maintained for 7 days, during which the medium was gently refreshed every two days to preserve the nutrient balance and remove debris or aberrant (irregular) spheroids. To improve spheroid uniformity, the plates were transferred to a rotary shaker incubator (60 rpm) on the second day of culture, which promoted even shear forces and homogeneous spheroid formation. Spheroid growth was monitored daily under an inverted microscope (Leica Microsystems, GmbH, Wetzlar, Germany), and the diameters were quantified using a calibrated eyepiece graticule (Carl Zeiss AG, Oberkochen, Germany) or ImageJ software for digital images. At least 30 spheroids per condition were measured to determine the mean spheroid diameter ± S.E.M.

### 4.10. Viability Assay

Cell viability was evaluated using crystal violet (Sigma-Aldrich; Cat. No. C0775) staining, as previously described by Feoktistova et al. (2016) [55], with minor modifications. Briefly, 5 × 10^3^ cells per well were seeded in 200 μL of complete culture medium into 96-well flat-bottom plates (Corning, USA) and allowed to adhere overnight at 37 °C in 5% CO_2_. At 0 and 72 h, the wells were gently washed twice with PBS to remove debris and non-adherent cells. Attached cells were fixed with 100% methanol for 10 min at room temperature and air-dried. Subsequently, the cells were stained with 0.05% (*w*/*v*) crystal violet solution (Sigma-Aldrich) for 15 min, followed by thorough washing with distilled water to remove excess dye and air-drying. To quantify cell viability, 33% (*v*/*v*) acetic acid was added to each well to solubilize the bound dye, and the optical density (*OD*) was measured at 570 nm using a microplate reader (Synergy HT; BioTek, Winooski, VT, USA). Relative cell viability (%) was calculated as follows:$$\mathrm{Cell}\mathrm{viability}=\frac{\mathrm{OD}72\;h}{\mathrm{OD}0\;h}\times 100$$

All treatments were performed in triplicate wells across three independent experiments, and the results were expressed as mean ± S.E.M. Representative images of the stained wells were captured using a digital camera at 1× magnification before quantification.

### 4.11. Clonogenicity Assay

BxPC-3, AsPC-1, and Capan-2 cells were seeded at a low density (300–800 cells/well, depending on the cell line) in 6-well plates and allowed to adhere overnight. The cells were then treated with N), PLP-Ev, or DRPs and incubated for 10–14 days until visible colonies (>50 cells) were formed. At each passage, parallel wells were fixed with 4% paraformaldehyde (Sigma-Aldrich; Cat. No. 158127) for 10 min and stained with 0.05% crystal violet to confirm colony formation. For serial clonogenicity assessment, colonies were trypsinized, resuspended as single-cell suspensions, and re-plated at clonal density under identical conditions for two additional passages (P2–P3). At the final passage, the colonies were fixed and stained as described above. Colony numbers were quantified using ImageJ software (NIH, USA) and expressed as the survival fraction relative to the Nt. The persistence of colony-forming ability across passages is interpreted as evidence of self-renewal potential. All experiments were performed in triplicate across three independent assays.

### 4.12. Western Blot Analysis

Total protein lysates from PTLs, PLPs-Ev, and DRPs cells were extracted using RIPA lysis buffer (50 mM Tris-HCl, pH 7.4; 150 mM NaCl; 1% NP-40; 0.1% SDS; 0.5% sodium deoxycholate) supplemented with protease and phosphatase inhibitor cocktails (Sigma-Aldrich, Cat. #P8340 and #P0044, St. Louis, MO, USA). Protein concentrations were quantified using the Pierce™ BCA Protein Assay Kit (Thermo Fisher Scientific, Cat. #23225, Waltham, MA, USA) according to the manufacturer’s instructions. Equivalent amounts of total protein (20–30 µg) were mixed with 4× Laemmli sample buffer (Bio-Rad, Cat. #1610747) containing 5% β-mercaptoethanol and denatured at 95 °C for 5 min. Proteins were resolved by SDS-PAGE on 12% polyacrylamide gels and transferred onto 0.2 µm nitrocellulose membranes (Bio-Rad, Cat. #1620112) using a Trans-Blot^®^ Turbo Transfer System (Bio-Rad Laboratories, Hercules, CA, USA). Membranes were blocked for 1 h at room temperature with Intercept^®^ TBS Blocking Buffer (LI-COR, Cat. #927-60001) and incubated overnight at 4 °C with primary antibodies diluted in blocking buffer containing 0.1% Tween-20. The following primary antibodies were used: Anti-DEFA1/3 antibody (mouse monoclonal, Abcam plc, Waltham, MA, USA, Cat. #ab97409, dilution 1:1000), anti-GAPDH antibody (rabbit monoclonal, Cell Signaling Technology, Danvers, MA, USA, Cat. #5174, dilution 1:5000) was used as the loading control. After washing, membranes were incubated for 1 h at room temperature with species-specific infrared secondary antibodies: Goat anti-mouse IgG (IRDye^®^ 680RD, LI-COR, Cat. #926-68070, 1:10 000), and goat anti-rabbit IgG (IRDye^®^ 800CW, LI-COR #926-32211, 1:10,000). Fluorescent signals were detected using an Odyssey^®^ CLx Infrared Imaging System (LI-COR Biosciences, Lincoln, NE, USA). Band intensities were quantified using Image Studio™ Lite, version 5.2 (LI-COR Biosciences, Lincoln, NE, USA), and DEFA1/3 protein content was normalized to GAPDH. All Western blot assays were performed in biological triplicates to ensure reproducibility. Full, uncropped Western blot images are shown in Appendix A.

### 4.13. Zebrafish Husbandry and Ethical Compliance

Adult zebrafish (*Danio rerio*) of the Tab-wik strain, originally provided by Dr. Ernesto Maldonado (Institute of Marine Sciences and Limnology, UNAM) and Dr. Francisco Carmona (Institute of Cellular Physiology, UNAM), were maintained under standard aquaculture conditions at 28.5 °C, with pH adjusted to 7.0–7.5, and a 14 h light/10 h dark cycle. Embryos were kept in embryo water (distilled water supplemented with 5 g/L Instant Ocean^®^ Salt and 7.5 mg/L NaHCO_3_).

General husbandry procedures for both adults and larvae followed established protocols detailed in The Zebrafish Book: A Guide for the Laboratory Use of Zebrafish (*Danio rerio*) (Westerfield, 2000) [56]. Experimental procedures were conducted according to international animal care standards, including the Institutional Animal Care and Use Committee (IACUC) guidelines (University of Oregon), and were approved by the Internal Committee for the Care and Use of Laboratory Animals (CICUAL; Comité Interno para el Cuidado y Uso de Animales de Laboratorio, INSTITUTO NACIONAL DE MEDICINA GENOMICA (INMEGEN); approval no. 427). For euthanasia, embryos older than 4 days post-fertilization (dpf) were sacrificed by immersion in chilled water (2–4 °C) until opercular movements ceased, followed by additional exposure to the same chilled water for several minutes, as recommended by the American Veterinary Medical Association (AVMA) Guidelines for the Euthanasia of Animals (2020) [57].

Animal care and monitoring. All procedures were designed to minimize the pain and distress. Embryos were anesthetized with 0.04% tricaine (MS-222; Sigma-Aldrich, Cat. No. E10521) during microinjection and was handled under a stereomicroscope to avoid mechanical stress. Embryos were monitored periodically during the 24 h post-injection period for signs of abnormal morphology, hemorrhage and death. Those presenting malformations or loss of viability were excluded from the analysis. No unexpected adverse events were observed. Humane endpoints were defined as death or severe malformation before imaging at 24 h post-injection, at which point the surviving embryos were imaged and subsequently maintained under standard conditions. Euthanasia was performed on larvae older than 4 dpf according to the AVMA guidelines.

### 4.14. Embryo Preparation, Microinjection, and Migration Assays

At 2 dpf, the embryos were manually dechorionated using Dumont No. 5 forceps and anesthetized in 0.04% tricaine (MS-222; Sigma-Aldrich, Cat. No. E10521). The embryos were positioned in 3% agarose-coated Petri dishes to ensure stability during injection. BxPC-3 pancreatic cancer cells pretreated with DRPs, PLPs, or left Nt were used for the xenotransplantation. Approximately 300 cells were injected into the yolk sac of each embryo. Each zebrafish embryo was considered an independent experimental unit. For each independent experiment, 20 embryos were allocated to each group (DRP, PLPs, and Nt), resulting in approximately 60 embryos per replicate and 300 embryos across all experiments. The chosen sample size followed previously validated zebrafish xenograft models (Martínez-López et al., 2021) [58,59] that demonstrated consistent tumor dissemination variability within this range, ensuring reproducibility and statistical reliability without requiring a formal a priori power calculation. Cells were pre-labeled with CellTracker™ Red CMTPX (Thermo Fisher Scientific, Cat. No. C34552), were resuspended at 1 × 10^3^ cells/10 μL, and ~300 cells were injected into the yolk sac of each embryo, under blinded conditions, following xenotransplantation procedures adapted from Martínez-López et al. 2021 [58,59]. Borosilicate glass capillaries (10 cm length, 1.0 mm outer diameter (OD), 0.58 mm inner diameter (ID), without filament; Sutter Instrument, Novato, CA, USA) were pulled using a vertical pipette puller (Model 51210, Stoelting Co., Wood Dale, IL, USA) and trimmed 0.8 cm from the shoulder using Dumont No. 5 forceps. Injections were performed using a Pneumatic Pico-Liter Injector (PLI-100A; Warner Instruments, Holliston, MA, USA).

Post-injection, the embryos were rinsed with embryo water and incubated at 34 °C in Petri dishes containing sufficient embryo water, with a density of ≤30 embryos per dish. The inclusion criteria were defined a priori as (i) viable 2 dpf embryos and (ii) successful xenotransplantation verified immediately after injection (T0) by the presence of fluorescently labeled tumor cells at the yolk sac injection site. Embryos remained eligible for analysis regardless of whether fluorescent cells were confined to the yolk sac or had already disseminated at 24 h post-injection; migration was treated as an outcome. The exclusion criteria (predefined) were technical failure of injection (off-target delivery or widespread leakage at T0), severe malformation or hemorrhage, or death prior to imaging at 24 h. Each experimental group included exactly 20 embryos (*n* = 20 per group, per replicate); the exact *n* used in each analysis is reported in the corresponding figure legend. At 24 h post-injection, the embryos were examined in a blinded manner using a Zeiss epifluorescence microscope equipped with a Canon digital camera. Images were processed using FIJI–ImageJ software, and the number of fluorescent cells detected in the posterior trunk and tail (beyond the yolk sac) was quantified [58].

Randomization and control of confounders. Randomization was performed according to the treatment sequence (temporal alternation) during the injection process. Embryos from mixed clutches were selected without distinction and injected sequentially in alternating order across the treatment groups (DRP, PLP, and Nt) to avoid time-dependent or operator bias. Each experimental day included all three groups to minimize batch effects. All embryos were maintained under identical conditions (temperature, density ≤30 embryos per dish, and 14/10 h light/dark cycle). The plate positions within the incubator were rotated daily to prevent location bias. Outcome assessment was performed 24 h post-injection under blinded conditions. The operator was aware of the treatment allocation during the injection; however, outcome evaluation was performed using coded image files to reduce observer bias.

Outcome measures. The primary outcome measure was the number of fluorescent BxPC-3 cells that migrated beyond the yolk sac 24 h post-injection, quantified using FIJI–ImageJ. This variable directly assessed the pro-tumorigenic effects of DRPs in vivo. The secondary outcome measures included the fluorescent area corresponding to disseminated cells and the percentage of embryos showing successful xenografts. These measures were selected based on previous zebrafish xenotransplantation studies that evaluated tumor dissemination and invasion dynamics (Martínez-López et al., 2021) [32]. The sample size (*n* = 20 embryos per group per replicate) was determined based on this primary outcome.

### 4.15. Transcriptomic Analyses and Survival Modeling

RNA-seq expression data and associated clinical annotations for PDAC were obtained from TCGA-PAAD via the Genomic Data Commons (GDC) Data Portal (https://portal.gdc.cancer.gov/ accessed on 1 March 2025). Raw counts were normalized to CPM and log_2_-transformed using the limma/voom pipeline implemented in R (v4.3.0). For survival analyses, patients were stratified into quartiles based on *DEFA1/3* expression, and OS was compared using Kaplan–Meier curves generated with the survival version 3.7-0 and survminer version 0.4.9 R packages. To evaluate potential interactions with circulating platelet levels, patients were further subdivided into four categories (high/low *DEFA1/3* × high/low platelets) based on the median values of both variables. Log-rank tests were used to assess statistical significance. All analyses were conducted in R using publicly available TCGA data, following established computational reproducibility guidelines.

### 4.16. Single-Sample Gene Set Enrichment Analysis

To explore functional associations, ssGSEA was performed using the Gene Set Variation Analysis (GSVA) R package (v3.21). Curated gene sets related to platelet activation, neutrophil infiltration, and EMT were used as reference pathways [60]. Spearman’s rank correlation was applied to assess associations between *DEFA1/3* expression and ssGSEA enrichment scores across samples. Correlation strength was quantified using the rho coefficient, and *p*-values were calculated to determine the significance. Scatter plots with linear regression lines were generated using the ggplot2 R package version 3.5.1, with rho and *p*-values indicated in each panel [61]. All analyses were conducted in R (v4.3.0) using publicly available TCGA-PDAC data.

### 4.17. Correlation Analyses with Aggressiveness-Related Genes

To further explore the molecular programs linked to PDAC aggressiveness, we selected a panel of genes previously reported to stratify tumor subtypes and progression (*SPARC*, *GATA6*, *KDM6A*, and *EGR1*). Transcriptomic scores for invasion (MMPs), motility (integrins), and EMT-like programs were computed using curated hallmark and Gene Ontology (GO) gene sets. Associations between gene expression and program scores were tested using Spearman’s correlation, and regression plots were visualized with confidence intervals [62].

To further explore the molecular programs associated with PDAC aggressiveness, we analyzed a curated panel of genes previously reported to stratify pancreatic tumor subtypes and progression (*SPARC*, *GATA6*, *KDM6A*, and *EGR1*) [63]. Transcriptomic activity scores were calculated for invasion (MMP-related), motility (integrin-related), and EMT-like programs using hallmark and GO gene sets from the Molecular Signatures Database (MSigDB v7.5). Associations between gene expression and program scores were evaluated using Spearman’s rank correlation, and results were visualized as regression plots with 95% confidence intervals generated in R (v4.3.0).

### 4.18. Statistical Analysis

Data are presented as mean ± S. E. M. from a minimum of three independent experiments, each conducted in triplicate. Group comparisons were performed using one-way analysis of variance (ANOVA), followed by Tukey’s post hoc test to assess pairwise differences. Statistical significance was set at *p* < 0.01. All statistical analyses and graphical representations were performed using GraphPad Prism software version 9.0 (GraphPad Software, San Diego, CA, USA).

## 5. Conclusions

This study provides compelling evidence that α-defensins, specifically *DEFA1/3*, play a previously underappreciated role in PDAC progression. While most prior research has centered on β-defensins, our findings highlight the importance of α-defensins as both biomarkers and functional modulators of tumor aggressiveness in this malignancy.

By developing a renewable in vitro platelet production model, we demonstrated that platelet-derived α-defensins significantly enhance cancer cell viability, migration, clonogenicity, and spheroid formation. This system faithfully replicates the behavior of platelets isolated from healthy donors, reinforcing its translational utility.

Moreover, our in vivo zebrafish xenograft assays confirmed that *DEFA1/3*-enriched platelets promote tumor growth and dissemination in a complex biological environment. At the molecular level, DRPs selectively induced the expression of aggressiveness-associated genes, including *SPARC, KDM6A* and *GATA6*. Patient data analyses linked high *DEFA1/3* expression with poor prognosis, immune infiltration, and EMT activation.

## Figures and Tables

**Figure 1 ijms-26-10898-f001:**
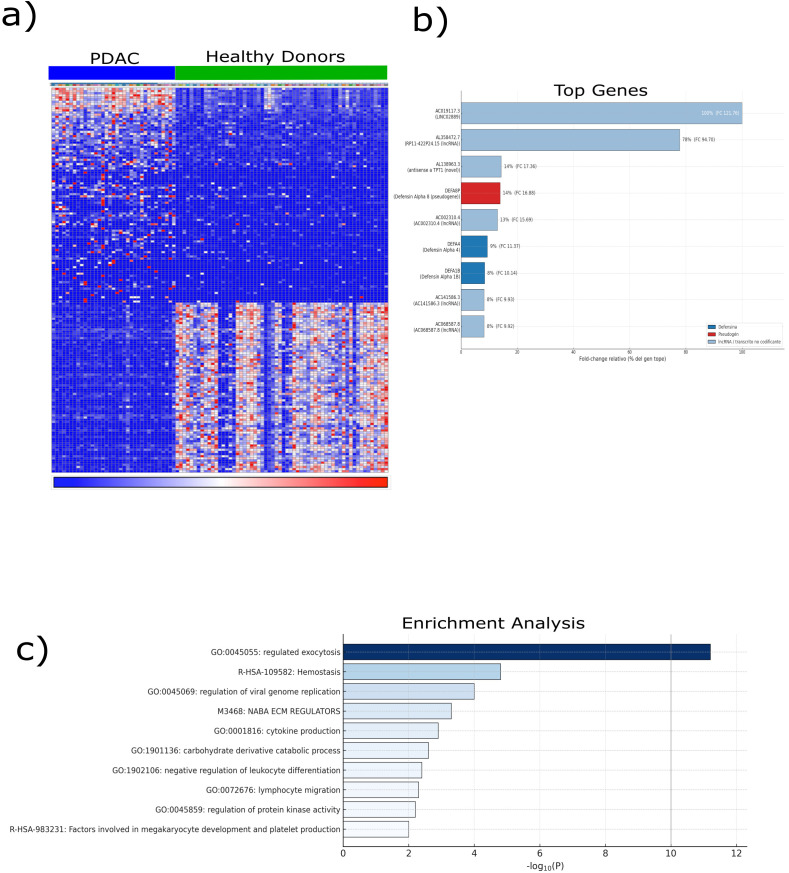
Bioinformatic characterization of transcriptomic alterations in pancreatic cancer. Transcriptomic data were obtained from the GSE68086 dataset, which comprised RNA-seq profiles of platelets from patients with pancreatic ductal adenocarcinoma (PDAC) (*n* = 35) and healthy controls (*n* = 55). (**a**) Unsupervised hierarchical clustering of differentially expressed genes (|log_2_FC| ≥ 1, adjusted *p* < 0.05, Benjamini–Hochberg correction) was performed using the limma R package (v4.3), revealing clear segregation between the PDAC and control platelet transcriptomes. The heatmap color scale represents normalized gene expression levels, where red indicates upregulation and blue indicates downregulation relative to the mean expression across samples. (**b**) Top ten overexpressed genes in PDAC platelets are shown, highlighting defensin family members (DEFA4, DEFA1B, and DEFA8P) for their strong upregulation and established roles in immune defense and epithelial barrier integrity. (**c**) Gene Set Enrichment Analysis (GSEA) was performed using the ClusterProfiler (v4.10.1) and fgsea (v1.28.0) R packages with the MSigDB hallmark collection. The normalized enrichment score (NES) reflects the magnitude and direction of gene set over-representation. Nominal *p*-values and FDR q-values were derived from 10,000 phenotype-based permutations. Collectively, these analyses indicate that the platelet transcriptomes of patients with PDAC exhibit immune and secretory reprogramming, supporting the concept of tumor-educated platelets as active modulators of cancer progression. Data are presented as mean expression ± S.D., and statistical significance was defined as adjusted *p* < 0.05. Abbreviations: PDAC, pancreatic ductal adenocarcinoma; GSEA, Gene Set Enrichment Analysis; NES, normalized enrichment score; RNA-seq, RNA sequencing; DEFA, defensin alpha.

**Figure 2 ijms-26-10898-f002:**
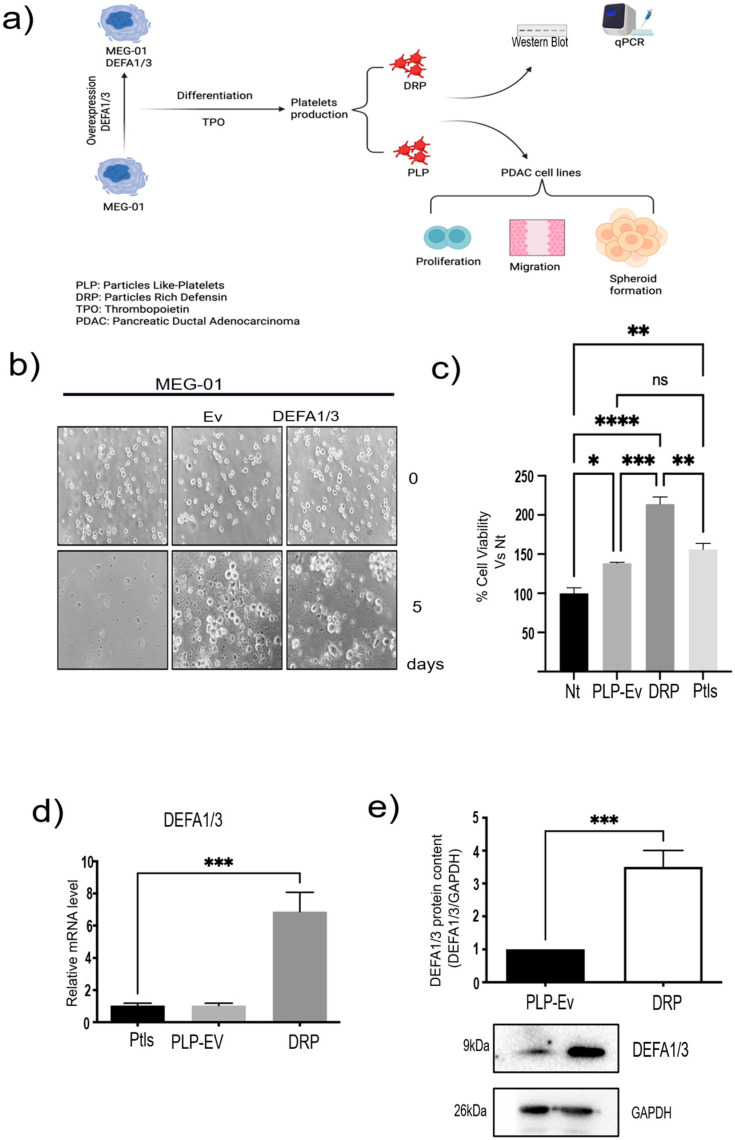
Generation and validation of DRPs and PLPs from MEG-01 cells. (**a**) Schematic overview of the experimental workflow. MEG-01 cells were transduced with lentiviral vectors encoding DEFA1/3 or an empty vector (EV) control and subsequently differentiated to produce platelet-like particles (PLPs) and defensin-rich platelets (DRPs). These particles were used for molecular characterization (qPCR, Western Blot) and functional assays in pancreatic ductal adenocarcinoma (PDAC) cell lines, including proliferation, migration, and spheroid formation. (**b**) To generate DRPs, MEG-01 cells were genetically modified to overexpress DEFA1/3 by lentiviral transduction. Viral particles were produced by co-transfecting 293T cells with pLV-IRES-DEFA1/3 (donor vector), pLJM1-GFP (transduction control), psPAX2 (packaging plasmid), and pCMV-VSV-G (envelope protein). Viral supernatants were collected every 12 h for three days and used to transduce MEG-01 cells. Representative micrographs show platelet-like particles released by MEG-01-EV and MEG-01-DEFA1/3 cells (scale bar = 10 µm). (**c**) Cell viability assays were performed after 7 days of co-culture of PDAC cells with PLPs-Ev, DRPs, or peripheral blood-derived platelets (PLTs) to evaluate potential cytotoxicity. (**d**) Relative *DEFA1/3* mRNA levels were quantified using qPCR in parental MEG-01, MEG-01-EV, and MEG-01-DEFA1/3 cells. (**e**) Protein expression was confirmed by Western blotting, and densitometric quantification was normalized to GAPDH. Each bar represents the mean ± standard error of the mean (S.E.M.) from at least four independent experiments, each performed in triplicate. Statistical significance was determined using one-way ANOVA followed by Tukey’s post hoc test, with significance thresholds defined as * *p* ≤ 0.05, ** *p* ≤ 0.01, *** *p* ≤ 0.001, **** *p* ≤ 0.0001 and ns = not significant. This model provides a controlled in vitro platform to study how platelet-derived molecules, such as *DEFA1/3*, influence pancreatic cancer cell behavior, mimicking the functional interactions observed in circulating platelets in patients. Abbreviations: PDAC, pancreatic ductal adenocarcinoma; *DEFA1/3*, Defensin Alpha 1/3; DRPs, defensin-rich platelets; PLPs, platelet-like particles; EV, empty vector; PLTs, platelets; qPCR, quantitative polymerase chain reaction; WB, Western blotting; GAPDH, glyceraldehyde 3-phosphate dehydrogenase; Nt, non-treated control; S.E.M., standard error of the mean.

**Figure 3 ijms-26-10898-f003:**
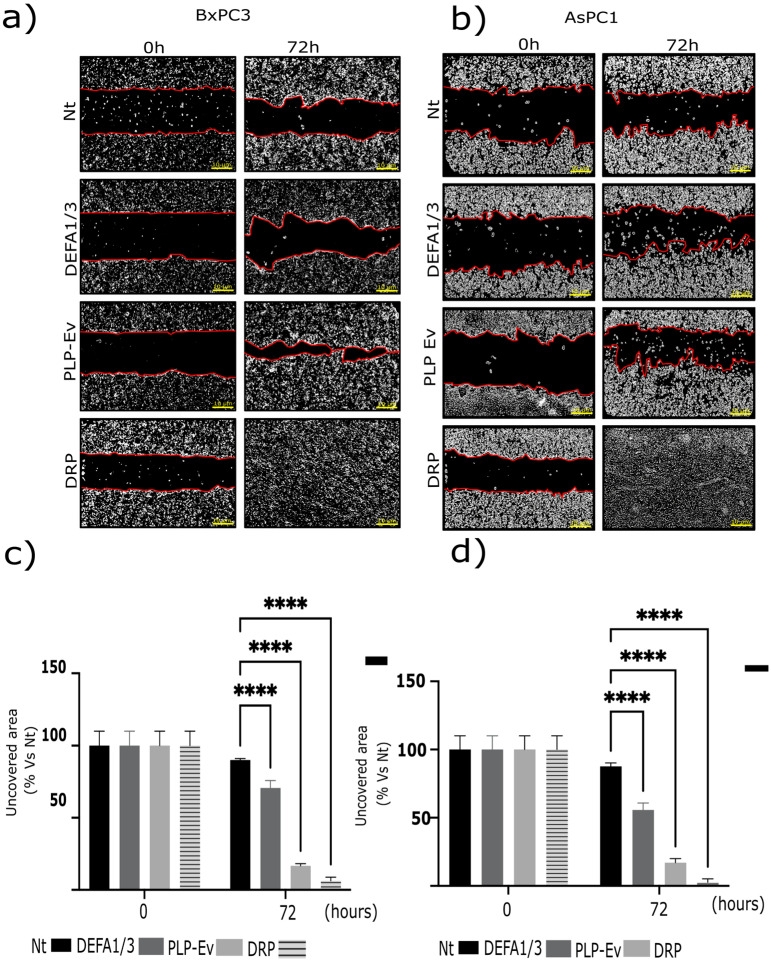
DRP promotes the migratory capacity of pancreatic cancer cell lines. The effects of defensin-rich platelets (DRPs) and recombinant human DEFA1/3 on cell migration were evaluated in AsPC-1 and BxPC-3 pancreatic cancer cells using a wound-healing assay. Cells were seeded in Ibidi^®^ culture-insert 2-well chambers and grown until they reached confluence. The inserts were then removed to create a defined cell-free gap (“wound”), and wound closure was monitored at baseline (T_0_) and after 72 h of incubation in a serum-free medium supplemented with 5 µM cytosine arabinoside (Ara-C) to inhibit proliferation. (**a**,**b**) Representative phase-contrast images showing wound closure at T_0_ and 72 h for BxPC-3 (**a**) and AsPC-1 (**b**) cells treated with non-treated control (Nt), recombinant DEFA1/3 (100 ng/mL), PLP-Ev, or DRPs. Red lines indicate the wound edges used to delineate the uncovered area for quantification. Scale bar = 10 µm. (**c**,**d**) Quantification of wound closure, expressed as the percentage of uncovered wound area relative to Nt at T_0_ for BxPC-3 (**c**) and AsPC-1 (**d**) cells. Patterned gray bars represent DRP-treated conditions in panel (**c**,**d**). Migration was quantified using the ImageJ software version 1.53t (NIH). Data represent the mean ± S.E.M. of at least three independent biological replicates, each performed in triplicate. Statistical analysis was performed using one-way ANOVA followed by Tukey’s post hoc test, comparing each treatment between T_0_ and 72 h. Significance levels: **** *p* ≤ 0.0001 vs. Nt. These findings indicate that defensin-enriched platelet-like particles promote cancer cell motility independently of cell proliferation, which is a hallmark of platelet-mediated tumor dissemination. Abbreviations: DRPs, defensin-rich platelets; PLPs, platelet-like particles; PLP-EV, empty vector-derived PLPs; DEFA1/3, Defensin Alpha 1/3; Nt, non-treated control; Ara-C, cytosine β-D-arabinofuranoside; PDAC, pancreatic ductal adenocarcinoma; S.E.M., standard error of the mean.

**Figure 4 ijms-26-10898-f004:**
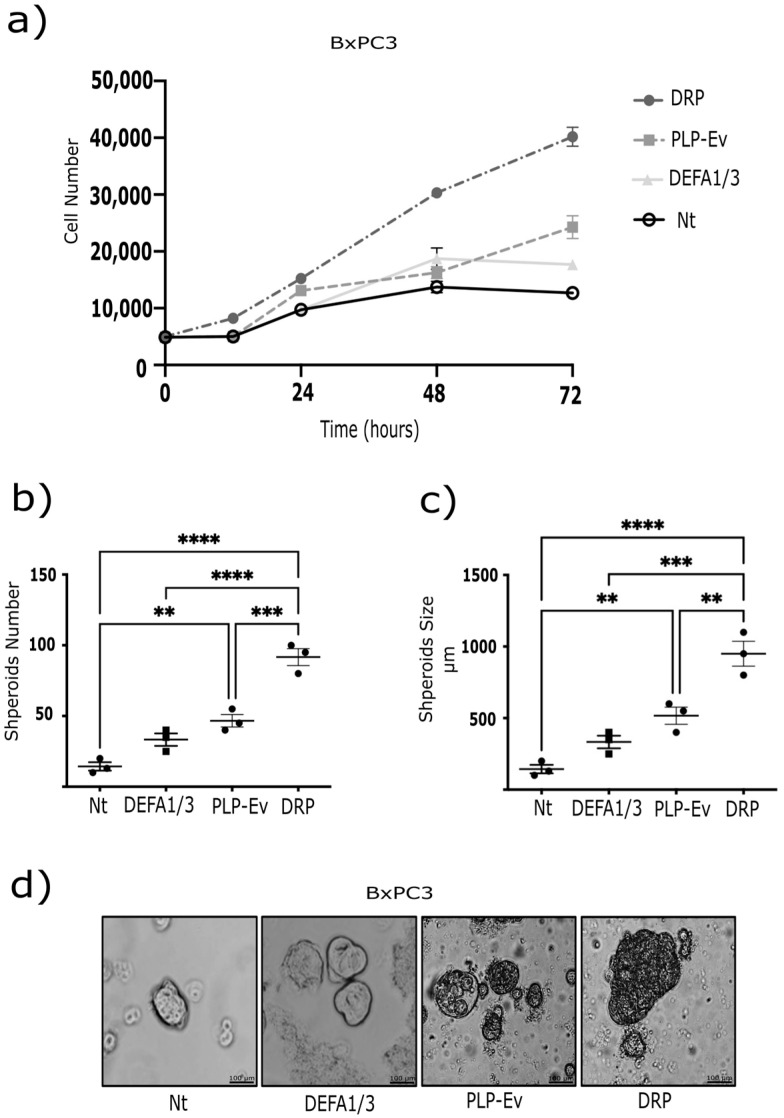
DRP enhances spheroid formation and proliferation of pancreatic cancer cells. (**a**) BxPC-3 cell proliferation was evaluated using the CCK-8 assay. Each point represents the mean ± S.E.M. of at least four independent experiments performed in triplicates. Statistical differences were assessed using one-way ANOVA followed by Tukey’s post hoc test (** *p* ≤ 0.01, *** *p* ≤ 0.00 and **** *p* ≤ 0.0001 vs. Nt). (**b**) BxPC-3 cells grown as monolayers were seeded in Petri dishes coated with a thin layer of 2% agarose and cultured for one week. Spheroids were then transferred to a rotary incubator at 37 °C in Leibovitz’s L-15 medium without fetal bovine serum (FBS) and treated with DRPs or PLPs-Ev. (**c**) The spheroid number and size were quantified using ImageJ. Each data point represents an independent experiment; data are expressed as median ± S.E.M. from at least three independent experiments conducted in triplicate. *p* ≤ 0.01 vs. Nt, by ANOVA with Tukey’s post hoc test. (**d**) Representative phase-contrast micrographs of spheroids (scale bar = 100 µm). The observed increase in spheroid size and number suggests that DRPs reinforce three-dimensional tumor growth and survival, consistent with the pro-tumorigenic role of platelet-derived factors in PDAC. Abbreviations: DRPs, defensin-rich platelets; PLPs, platelet-like particles; PLPs-EV, empty vector-derived PLPs; PDAC, pancreatic ductal adenocarcinoma; Nt, non-treated control; CCK-8, Cell Counting Kit-8; FBS, fetal bovine serum; S.E.M., standard error of the mean.

**Figure 5 ijms-26-10898-f005:**
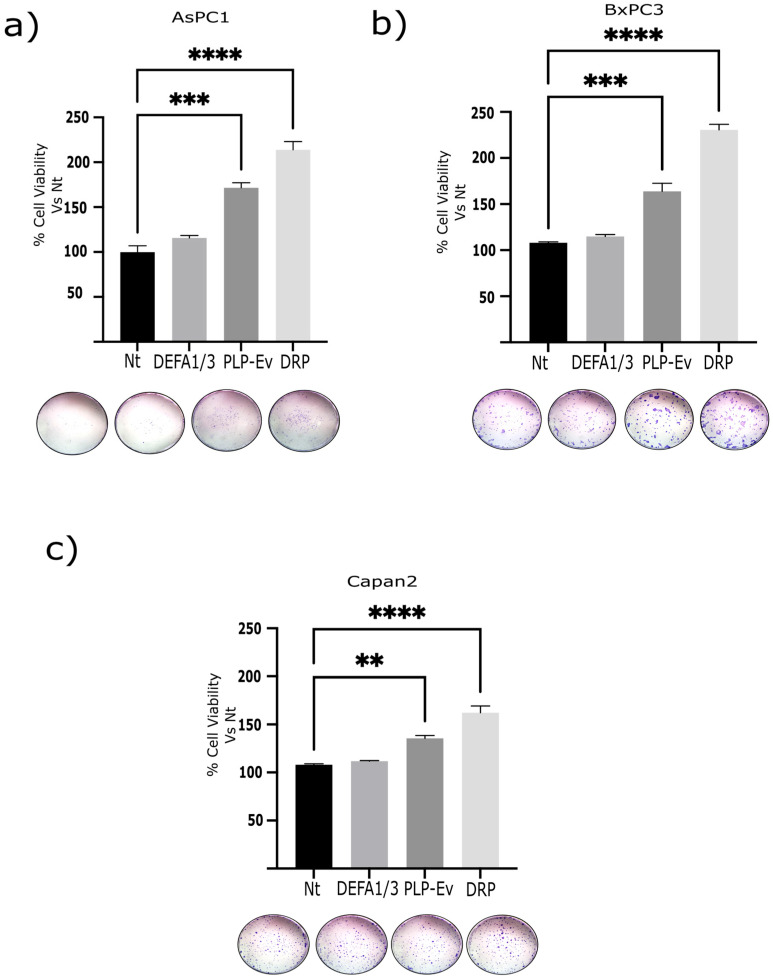
Serial clonogenicity assays demonstrated enhanced self-renewal capacity upon DRP treatment. Schematic representation of the serial clonogenic assay. PDAC cells (BxPC-3, AsPC-1, and Capan-2) were seeded at low density, treated with Nt, DEFA1/3, PLP-Ev, or DRPs, and allowed to form colonies for 10–14 days. Colonies (>50 cells) were fixed, stained with 0.05% crystal violet, trypsinized, and re-plated at clonal density for two additional passages (P2–P3) to evaluate self-renewal capacity. Colonies were then trypsinized, re-plated at clonal density, and cultured for an additional 10–14 days post seeding. (**a**–**c**) Quantification of colony numbers across serial passages. DRP-treated cells retained significantly higher clonogenic capacity across passages compared to the Nt and PLP-Ev groups. Representative crystal violet–stained images of colonies at passage 3. Data are presented as mean ± S.E.M. from three independent experiments, each performed in triplicate. Statistical comparisons were performed using one-way ANOVA followed by Tukey’s post hoc test, with significance thresholds defined as ** *p* ≤ 0.01, *** *p* ≤ 0.001, and **** *p* ≤ 0.0001. Representative images of crystal violet–stained colonies are shown for passage 3. Representative images of crystal violet–stained colonies are shown for passage 3. Representative images captured using a stereomicroscope (non-calibrated magnification). Scale is not shown due to variability in optical zoom. The enhanced clonogenic potential following DRP exposure suggests that platelet-derived defensins may contribute to tumor cell persistence and regenerative capacity. Abbreviations: PDAC, pancreatic ductal adenocarcinoma; DRPs, defensin-rich platelets; PLPs, platelet-like particles; PLPs-EV, empty vector–derived PLPs; DEFA1/3, Defensin Alpha 1/3; Nt, non-treated control; S.E.M., standard error of the mean.

**Figure 6 ijms-26-10898-f006:**
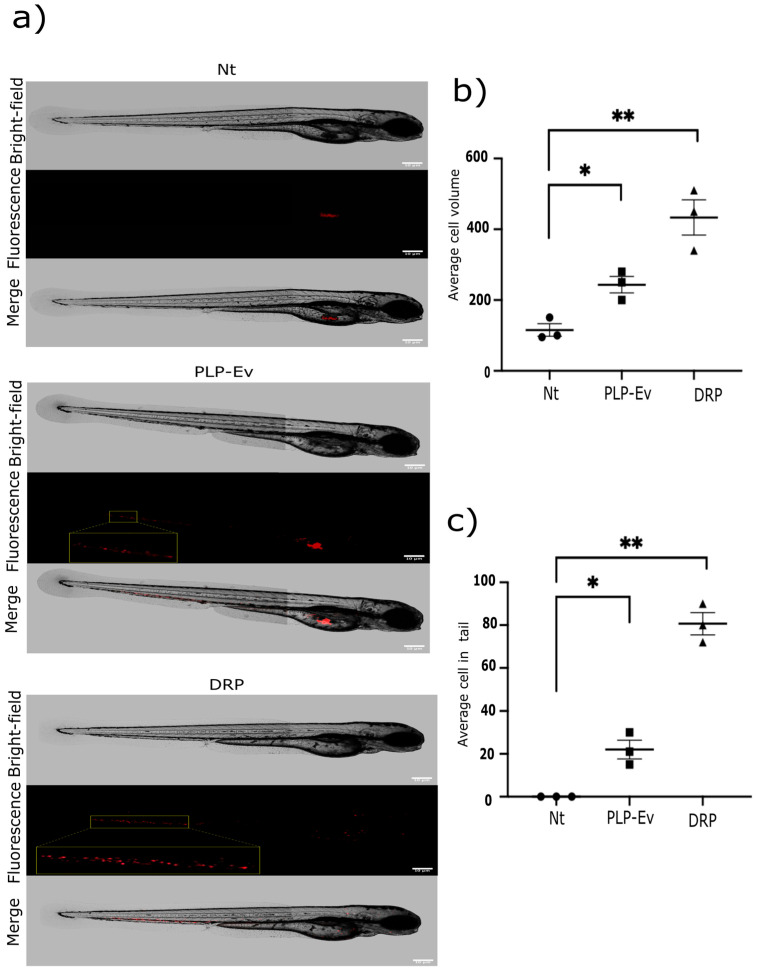
DRP enhances in vivo migration in a zebrafish model. In vivo migration assay of BxPC-3 cells in zebrafish embryos. An in vivo migration assay of BxPC-3 pancreatic cancer cells was conducted using zebrafish embryos as a xenograft model. (**a**) Representative images of 2-day-old embryos at 24 h post-injection (hpi) with ~300 BxPC-3 cells previously treated with DRPs or PLP-Ev. Images are shown in bright-field (top), epifluorescence (middle), and merged (bottom) views. Red fluorescence indicates the location of human pancreatic cancer cells within the embryos. Scale bar = 100 µm. (**b**) Quantification of the total fluorescent area (spheroid volume) and (**c**) the number of disseminated fluorescent cells located outside the yolk sac were measured using FIJI–ImageJ software version 1.54p. Data are presented as the mean ± SD from three independent experiments, each including at least 20 embryos per condition. Statistical significance was determined using one-way ANOVA followed by Tukey’s post hoc test, with significance thresholds defined as * *p* ≤ 0.05, ** *p* ≤ 0.01. Zebrafish xenografts provide a dynamic model for visualizing tumor–platelet interactions in vivo, and the results confirmed that DRP-treated cells displayed increased invasiveness and dissemination potential. Abbreviations: DRPs, defensin-rich platelets; PLPs, platelet-like particles; PLPs-EV, empty vector–derived PLPs; PDAC, pancreatic ductal adenocarcinoma; Nt, non-treated control; hpi, hours post-injection; S.D., standard deviation.

**Figure 7 ijms-26-10898-f007:**
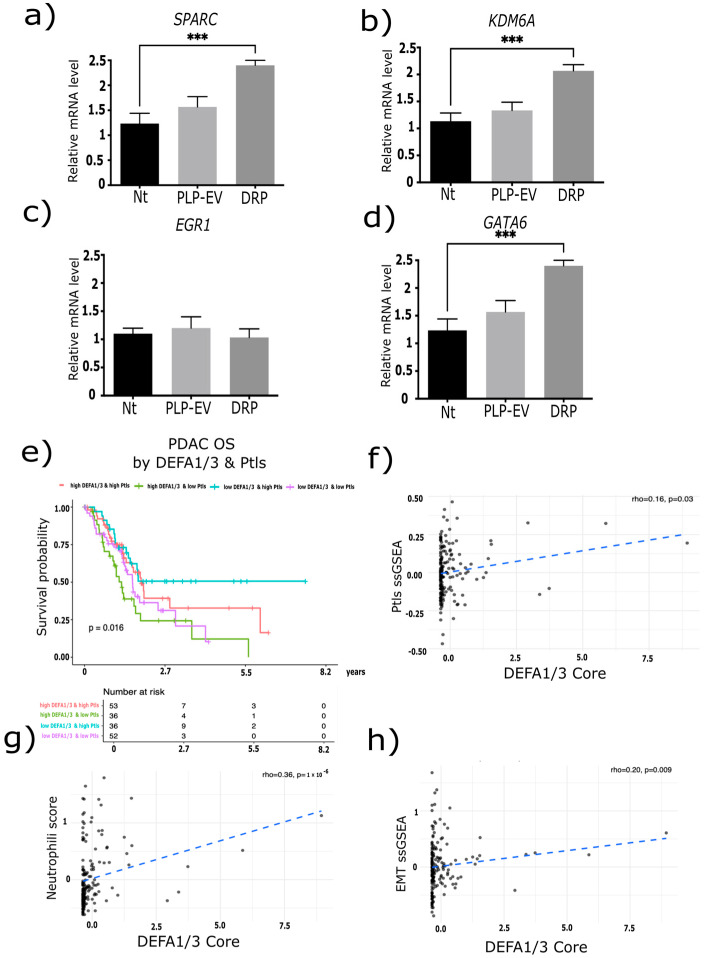
DRP modulates the gene expression in pancreatic cancer cells. BxPC-3 pancreatic cancer cells were treated with defensin-rich platelets (DRPs) or control platelet-like particles (PLPs-Ev), and the mRNA expression levels of (**a**) *SPARC*, (**b**) *KDM6A*, (**c**) *EGR1*, and (**d**) *GATA6* were quantified using RT-qPCR. Data are presented as relative mRNA expression normalized to GAPDH, expressed as the mean ± S.E.M. from three independent experiments, each performed in triplicate. Statistical analysis was conducted using one-way ANOVA followed by Tukey’s post hoc test, with significance thresholds defined as *** *p* ≤ 0.001. (**e**) Kaplan–Meier survival analysis of overall survival (OS) in patients with pancreatic ductal adenocarcinoma (PDAC) from the TCGA-PAAD cohort (*n* = 177), stratified by high/low DEFA1/3 expression and high/low platelet counts. Patients with low *DEFA1/3* and high platelet levels showed the most favorable outcomes, whereas those with high *DEFA1/3* and low platelet levels had the poorest OS (log-rank test, *p* = 0.016). (**f**–**h**) the DEFA1/3 expression showed positive correlations with (**f**) platelet-related pathways (rho = 0.16, *p* = 0.03), (**g**) neutrophil infiltration (rho = 0.36, *p* = 1 × 10^−6^), (**h**) and epithelial–mesenchymal transition (EMT) activation (rho = 0.20, *p* = 0.009). The blue dashed line represents the linear regression fit between DEFA1/3 expression and each indicated pathway signature. Together, these molecular and clinical analyses revealed that *DEFA1/3* expression is associated with pathways linked to platelet activation, immune cell infiltration, and EMT, integrating platelet-derived defensins into the transcriptional programs driving PDAC aggressiveness. Abbreviations: DRPs, defensin-rich platelets; PLPs, platelet-like particles; PLPs-EV, empty vector–derived PLPs; PDAC, pancreatic ductal adenocarcinoma; DEFA1/3, Defensin Alpha 1/3; OS, overall survival; TCGA, The Cancer Genome Atlas; ssGSEA, single-sample Gene Set Enrichment Analysis; EMT, epithelial–mesenchymal transition; Nt, non-treated control; S.E.M., standard error of the mean.

## Data Availability

The data presented in this study are available upon request from the corresponding author if you want to partner or contribute to the project. The data are not publicly available because of institutional review board politics.

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
