# Peer review of "Defensin-Rich Platelets Drive Pro-Tumorigenic Programs in Pancreatic Adenocarcinoma"

_ijms, 2025, doi:10.3390/ijms262210898_

Round 1
Reviewer 1 Report
Comments and Suggestions for Authors
In this manuscript, González-Ruiz et al. present a well-designed study showing that platelets rich in defensins (DEFA1/3) can promote the aggressive nature of pancreatic ductal adenocarcinoma (PDAC). Using data analysis, cell-based experiments, and animal models, the authors demonstrate how these platelet factors support cancer growth and spread. However, I have a few questions that need clarification in some parts.
1) The introduction looks fine, but it could better highlight why platelet defensins are understudied compared to other platelet factors.
2) In results section, adding more cohesive narrative linking each experiment to the study hypothesis would improve readability.
3) The discussion provides strong context, though a clearer distinction between correlation and functional implication would strengthen interpretation.
4) Figure legends could offer more explanatory detail to guide readers unfamiliar with platelet biology or PDAC models.
5) Simplify the methods portion in abstract section to emphasize the main findings more clearly.
6) The authors make an important contribution by integrating transcriptomic and in vivo data, but the transition between these sections feels abrupt and could flow more naturally.
7) Include the number of replicates in the figure captions to help the reader assess experimental robustness.
8) Add a graphical abstract.
Author Response
Response to Reviewers:
Reviewer 1
Comments and Suggestions for Authors
In this manuscript, González-Ruiz et al. present a well-designed study showing that platelets rich in defensins (DEFA1/3) can promote the aggressive nature of pancreatic ductal adenocarcinoma (PDAC). Using data analysis, cell-based experiments, and animal models, the authors demonstrate how these platelet factors support cancer growth and spread. However, I have a few questions that need clarification in some parts.
Response: We sincerely thank the reviewer for their positive assessment and constructive comments. We have carefully addressed each point to improve the clarity, coherence, and impact of the manuscript. Below are our detailed responses:
1)The introduction looks fine, but it could better highlight why platelet defensins are understudied compared to other platelet factors.
Response: We appreciate this valuable comment. We have expanded the final paragraph of the Introduction to emphasize that platelet defensins (DEFA1/3) have been historically overshadowed by better-characterized platelet factors such as TGF-β, PF4, and VEGF.
Introduction (page 2, lines 103–111)
´… The concept of tumor-educated platelets (TEPs) has emerged to describe platelets whose RNA and protein cargo are altered by tumor-derived signals. TEPs not only reflect the systemic influence of the tumor but also actively participate in shaping the tumor microenvironment through secreted mediators and intercellular communication(11,12). For instance, platelet-derived ITGA2B RNA has been identified as a robust biomarker for cancer detection, whereas platelet-secreted Transforming growth factor-beta (TGF-β) acts as a potent inducer of epithelial–mesenchymal transition (EMT), thereby directly promoting tumor progression(12,13).´
Introduction (page 2, lines 117–124)
´… While β-defensins have been extensively studied and θ-defensins are restricted to certain primates, platelet α-defensins remain comparatively understudied. This limited attention stems from both the technical challenges in their detection and their complex dual role: they can exert antimicrobial and immunomodulatory functions that may either suppress or enhance tumor progression, depending on the biological context (16,17). In contrast to well-characterized platelet factors such as TGF-β, Platelet factor 4 (PF4), and Vascular Endothelial Growth Factor (VEGF), the biological role of α-defensins in platelets remains largely unexplored. Importantly, platelets and mks have been shown to express Defensin Alpha 1/3 (DEFA1/3) transcripts and proteins, which localize to α-granules and are released upon activation(18), suggesting a previously underappreciated immunomodulatory role of platelet-derived α-defensins in the tumor milieu. Thus, it is conceivable that TEPs may carry defensins as part of their altered cargo, potentially contributing to tumor–platelet crosstalk and metastatic conditioning. ´
Introduction (pages 2–3, lines 133–221)
´… However, whether platelet-derived defensins, particularly DEFA1/3, contribute to PDAC progression has not been addressed. To explore this question in a physiologically relevant context, we employed zebrafish (Danio rerio) xenografts, a well-established model that enables real-time visualization of human cancer cell migration and reproduces conserved platelet–tumor interactions observed in humans. This study aimed to determine whether platelet-derived DEFA1/3 contributes to the aggressiveness of PDAC. Specifically, we hypothesized that platelets enriched in DEFA1/3 promote pro-tumorigenic and pro-migratory programs in PDAC cells. To test this hypothesis, we combined platelet transcriptomic re-analysis, in vitro functional assays, and in vivo zebrafish xenografts. Despite increasing evidence linking platelet-derived factors to tumor progression, the role of platelet defensins in pancreatic cancer biology remains largely unexplored. Here, we report that defensin-rich platelets (DRPs) promote proliferation, migration, and spheroid growth of pancreatic cancer cells, upregulate key aggressiveness-related genes, and are associated with adverse clinical outcomes, supporting the concept that platelet-derived DEFA1/3 represents a functional driver for PDAC progression and a candidate for translational exploration as a biomarker or therapeutic target... ´
2)In results section, adding more cohesive narrative linking each experiment to the study hypothesis would improve readability.
Response: We appreciate this valuable comment. We have revised the Results section to include clear transition statements linking each experimental block to the central study hypothesis. These narrative additions improve readability and emphasize the logical progression from transcriptomic findings to in vitro functional validation, in vivo confirmation, and clinical correlation.
Results (pages 3–4, lines 253-296)
´… Together, these findings provide an initial indication that platelet defensins, particularly DEFA family members, represent active components of platelet–tumor communication in PDAC, forming the basis for our subsequent functional validation. ´
Results (page 5, lines 328-329)
´Building on these transcriptomic findings, we next sought to determine whether platelet defensins exert functional effects on PDAC biology using a controlled in vitro model...´
Results (page 5, lines 328-329)
´…These results validated the transcriptomic observations and established a mechanistic framework to test whether DRPs modulate cancer cell migration and aggressiveness. ´
Results (pages 5, lines 361-362)
´ Given the established link between platelet activation and tumor dissemination, we assessed whether DRPs influence the migratory behavior of PDAC cells. …´
Results (page 7, lines 444-446)
´ As enhanced migration is often accompanied by increased proliferative and survival capacity, we investigated whether DRPs also promote PDAC cell growth and long-term expansion. …´
Results 2.5 (page 11, lines 644-646)
´ To verify whether the pro-tumorigenic effects of DRPs observed in vitro translate to a living system, we extended our analysis using a zebrafish xenograft model of human cancer …´
(Results 2.5 pages 11, lines 671-673)
…´ These in vivo data support our mechanistic findings, demonstrating that DEFA1/3-enriched platelets potentiate PDAC aggressiveness across biological scales. ´
Results 2.6 (page 13, lines 755-759)
´Having established that DEFA1/3-enriched platelets promote PDAC growth and dissemination in vivo, we next examined whether these effects were mirrored at the molecular level and reflected in patient-derived transcriptomic data from the TCGA database. To mechanistically link our experimental observations with clinical and transcriptomic evidence, we performed an integrated in vitro and in silico analysis of the data…´
Results 2.7 (pages 17, lines 952-955)
…Together with our functional results, these correlations suggest that DRPs rewire transcriptional programs associated with invasion and plasticity, supporting the central hypothesis that platelet-derived DEFA1/3 promotes pro-tumorigenic reprogramming in PDAC.
3)The discussion provides strong context, though a clearer distinction between correlation and functional implication would strengthen interpretation.
Response: We appreciate this insightful comment. We have revised the Discussion section to clarify the distinction between correlative observations and functional implications. Specifically, we added statements acknowledging that transcriptomic and clinical associations are inherently correlative and emphasized that causality was addressed through experimental validation using in vitro and in vivo models.
Discussion (pages 17, lines 966-969)
´… Although our analyses revealed strong associations between platelet-derived defensins and PDAC aggressiveness, these correlations alone do not establish causality. Therefore, we combined bioinformatic inference with functional experiments to directly assess the biological impact of these mutations. ´
Discussion (pages 18, lines 1016-1019)
´…These transcriptomic and survival associations are inherently correlative; however, their consistency with our mechanistic data reinforces the notion that platelet-derived defensins functionally contribute to PDAC progression…´
Discussion 3 (page 19, lines 1058-1060)
´…By integrating correlative and functional evidence, our study delineates how platelet defensins act as both biomarkers and active modulators of tumor behavior.
4)Figure legends could offer more explanatory detail to guide readers unfamiliar with platelet biology or PDAC models.
Response: We appreciate this helpful comment. We have expanded the Figure Legends to include explanatory context for readers less familiar with platelet biology and PDAC models. Specifically, we added short interpretative statements describing the biological meaning and relevance of each experiment (e.g., highlighting how DRPs model platelet–tumor interactions, or how zebrafish assays illustrate in vivo dissemination). These additions improve accessibility and guide interpretation without altering scientific content.
Figure Legend 1 (pages 4–5, lines 311-321)
´… Collectively, these analyses indicate that the platelet transcriptomes of patients with PDAC exhibit immune and secretory reprogramming, supporting the concept of tumor-educated platelets as active modulators of cancer progression.
. ´
Figure Legend 2 (page 7, lines 433-436)
´… This model provides a controlled in vitro platform to study how platelet-derived molecules, such as DEFA1/3, influence pancreatic cancer cell behavior, mimicking the functional interactions observed in circulating platelets in patients. ´
Figure Legend 3 (pages 9, lines 567-569)
´… These findings indicate that defensin-enriched platelet-like particles promote cancer cell motility independently of cell proliferation, a hallmark of platelet-mediated tumor dissemination. ´
Figure Legend 4 (page 11, lines 636-638)
´…The observed increase in spheroid size and number suggests that DRPs reinforce three-dimensional tumor growth and survival, consistent with a pro-tumorigenic role for platelet-derived factors in PDAC. ´
Figure Legend 5 (page 13, lines 746-748)
´… The enhanced clonogenic potential following DRP exposure suggests that platelet-derived defensins may contribute to tumor cell persistence and regenerative capacity. ´
Figure Legend 6 (page 15, lines 818-820)
´… Zebrafish xenografts provide a dynamic model to visualize tumor–platelet interactions in vivo, and the results confirm that DRP-treated cells display increased invasiveness and dissemination potential. ´
Figure Legend 7 (page 16, lines 859-862)
´… Together, these molecular and clinical analyses revealed that DEFA1/3 expression is associated with pathways linked to platelet activation, immune cell infiltration, and EMT, integrating platelet-derived defensins into the transcriptional programs driving PDAC aggressiveness. ´
5) Simplify the methods portion in abstract section to emphasize the main findings more clearly.
Response: We appreciate this valuable suggestion. We have simplified the methodological description in the Abstract to emphasize the main findings and overall significance of the study. The revised version now highlights the key experimental approaches concisely while focusing on the major results and their implications.
Abstract (page 1, lines 23-25)
Methods: We combined bioinformatic analysis of platelet transcriptomes with functional and in vivo zebrafish xenograft validation to investigate the impact of DEFA1/3 on PDAC aggressiveness.
Abstract (pages 1, lines 25-35)
´…Results: DEFA1/3 was significantly upregulated in PDAC-derived platelets. Defensin-enriched platelet-like particles (defensin-rich platelets, DRPs) and recombinant DEFA1/3 enhanced pancreatic cancer cell proliferation, migration, and three-dimensional growth in vitro and promoted tumor dissemination in zebrafish xenografts. Transcriptomic profiling revealed the upregulation of SPARC, KDM6A, and GATA6, whereas clinical data from The Cancer Genome Atlas (TCGA)-PDAC linked high DEFA1/3 expression to poor survival, increased immune infiltration, and activation of epithelial-mesenchymal transition (EMT). Conclusions: Platelet-derived DEFA1/3 acts as a functional modulator of PDAC progression, linking platelet granule content to tumor aggressiveness and highlighting a potential biomarker and therapeutic target within the platelet–tumor axis. ´
6) The authors make an important contribution by integrating transcriptomic and in vivo data, but the transition between these sections feels abrupt and could flow more naturally.
Response: We appreciate this constructive comment. We have improved the transition between the in vivo and transcriptomic sections to enhance narrative flow. Specifically, we added a bridging sentence clarifying that the transcriptomic and clinical analyses were performed to determine whether the in vivo phenotypes induced by DEFA1/3-rich platelets were reflected at the molecular and patient level. In addition, as part of the revisions described in point 2, we restructured the Results section to include connecting statements between experimental blocks, providing a more cohesive progression from transcriptomic analyses to in vitro and in vivo validation. Together, these adjustments substantially improve readability and ensure a smoother and more coherent integration of functional and transcriptomic data.
Results 2.6 (page 13, lines 754-757)
Having established that DEFA1/3-enriched platelets promote PDAC growth and dissemination in vivo, we next examined whether these effects were mirrored at the molecular level and reflected in patient-derived transcriptomic data from the TCGA database…´
7) Include the number of replicates in the figure captions to help the reader assess experimental robustness.
Response: We appreciate this helpful comment. We have revised all main and supplementary figure legends to specify the number of biological and technical replicates, as appropriate. In vitro assays now indicate that results are derived from at least three independent experiments (each performed in triplicate), and in vivo zebrafish experiments specify that each assay included a minimum of 20 embryos per condition. For bioinformatic and clinical analyses, the number of samples used (GSE68086: n = 35 PDAC, n = 55 controls; TCGA-PAAD: n = 177 tumor samples) has also been added. These revisions enhance the clarity, transparency, and reproducibility of our study.
(Figure legend 1, pages 4-5, lines 299-325).
(Figure legend 2, page 7, lines 413-440).
(Figure legend 3, pages 8-9, lines 546-572).
(Figure legend 4, pages 10-11, lines 613-641).
(Figure legend 5, pages 12-13, lines 724-751).
(Figure legend 6, pages 14-15, lines 780-823).
(Figure legend 7, pages 15-16, lines 845-867).
(Supplementary Figure 1, anexo1).
(Supplementary Figure 2, anexo1).
(Supplementary Figure 3, anexo1).
8) Add a graphical abstract.
Response: We appreciate the reviewer’s suggestion. In response, we have added a graphical abstract that summarizes the main findings and experimental design of the study. The new graphical abstract illustrates the generation of defensin-rich platelets (DRPs) from MEG-01 cells, their interaction with pancreatic cancer cells, and the resulting enhancement of migration, spheroid formation, and pro-tumorigenic signaling. This visual summary has been incorporated in accordance with the journal’s submission guidelines.

Reviewer 2 Report
Comments and Suggestions for Authors
The manuscript by González-Ruiz et al. entitled “Defensin-Rich Platelets Drive Pro-Tumorigenic Programs in Pancreatic Adenocarcinoma” addresses an important issue of unveiling the pathogenetic factors influencing the progression of pancreatic adenocarcinoma. The manuscript provides bioinformatic data, as well as in vitro and in vivo validation that defensins modulate the aggressiveness of pancreatic adenocarcinoma. Although multiple approaches are used to support the claim of the authors, the obtained findings not fully support the conclusions. Mechanisms through which defensins modulate the activity of tumor cells are not experimentally investigated. In general, the clarity and logical flow of the manuscript should be improved. It is difficult to follow it. Logical “bridges” connecting paragraphs are required for Introduction.
General comments:
- English level should be improved. The manuscript contains multiple grammatical and stylistic errors, typos, etc.
- In accordance with the HGNC (HUGO Gene Nomenclature Committee) guidelines, all gene symbols should be italicized.
Introduction:
- Introduction should be expanded by provided an in-depth background about the role of platelets and defensins in cancer
- Line 56. Introduce in a greater detail the concept of tumor-educated platelets and expand the information on its role in PDAC.
- Lines 56-60. The sentence is duplicated.
- Line 63. EMT abbreviation should be introduced here when the epithelial-mesenchymal transition is used for the first time in the manuscript
- Line 73. Abbreviation Mks for megakaryocytes should be introduced above when the term is mentioned for the first time in the manuscript text. Double check the abbreviations in the entire manuscript. Some abbreviations are not provided in full (PTLS, etc.)
Materials and methods:
- Protocols are not always sufficient enough to ensure reproducibility of this study. Expand them by providing detailed step-by-step instructions
- Healthy donors used as a source of PTLS should be characterized (number, age, sex, etc.)
- Any commercialized kits, reagents, instruments, software, antibodies, etc. used in the research, should be provided with their full name, along with the information of the manufacturers/suppliers/software details (Company Name, catalog number, City, Province/State, Country).
- Equipment used is not always described
- Provide primer sequences used for PCR
- No description of WB used
Results:
- Scale bars are not shown
- WB images should be presented uncut. Provide original WB images
- Figure captions should contain the list of abbreviations
- Figure 3. Figure caption should be significantly expanded to represent “independent” sources of the information. They should contain the description of what and how was measured, key findings, statistical analysis used, the way numerical data are presented (e.g., mean and SD, number of replicates, etc. Quality of images is insufficient. Scale bar is not visible. In general, captions are often not sufficient to understand what is shown.
- Precise p values should be mentioned in Figures
- English level should be improved. The manuscript contains multiple grammatical and stylistic errors, typos, etc.
Author Response
Reviewer 2
General comments:
- English level should be improved. The manuscript contains multiple grammatical and stylistic errors, typos, etc.
Response: We sincerely thank the reviewer for this valuable comment. We have carefully revised the entire manuscript to improve the English language, grammar, and overall readability. All sections were edited for clarity, conciseness, and consistency in scientific style.
- In accordance with the HGNC (HUGO Gene Nomenclature Committee) guidelines, all gene symbols should be italicized.
Response: We thank the reviewer for this observation. All gene symbols have been revised and formatted in accordance with HGNC (HUGO Gene Nomenclature Committee) guidelines, ensuring that they appear italicized throughout the manuscript, including text, figures, and legends.
Introduction:
- Introduction should be expanded by provided an in-depth background about the role of platelets and defensins in cancer.
Response: We appreciate this valuable suggestion. The Introduction section has been expanded to provide a more comprehensive background on the role of platelets and defensins in cancer biology. We now include an updated discussion of platelet-mediated tumor promotion mechanisms (e.g., TGF-β, PF4, VEGF pathways) and summarize the emerging evidence linking α-defensins to immune modulation and tumor progression across different cancer types, with emphasis on pancreatic cancer.
Introduction (page 2, lines 103-111)
´… The concept of tumor-educated platelets (TEPs) has emerged to describe platelets whose RNA and protein cargo are altered by tumor-derived signals. TEPs not only reflect the systemic influence of the tumor but also actively participate in shaping the tumor microenvironment through secreted mediators and intercellular communication(11,12). For instance, platelet-derived ITGA2B RNA has been identified as a robust biomarker for cancer detection, whereas platelet-secreted Transforming growth factor-beta (TGF-β) acts as a potent inducer of epithelial–mesenchymal transition (EMT), thereby directly promoting tumor progression(12,13).´
Introduction (page 2, lines 117–124)
´… While β-defensins have been extensively studied and θ-defensins are restricted to certain primates, platelet α-defensins remain comparatively understudied. This limited attention stems from both the technical challenges in their detection and their complex dual role: they can exert antimicrobial and immunomodulatory functions that may either suppress or enhance tumor progression, depending on the biological context (16,17). In contrast to well-characterized platelet factors such as TGF-β, Platelet factor 4 (PF4), and Vascular Endothelial Growth Factor (VEGF), the biological role of α-defensins in platelets remains largely unexplored. Importantly, platelets and mks have been shown to express Defensin Alpha 1/3 (DEFA1/3) transcripts and proteins, which localize to α-granules and are released upon activation(18), suggesting a previously underappreciated immunomodulatory role of platelet-derived α-defensins in the tumor milieu. Thus, it is conceivable that TEPs may carry defensins as part of their altered cargo, potentially contributing to tumor–platelet crosstalk and metastatic conditioning. ´
Introduction (pages 2–3, lines 133–221)
´… However, whether platelet-derived defensins, particularly DEFA1/3, contribute to PDAC progression has not been addressed. To explore this question in a physiologically relevant context, we employed zebrafish (Danio rerio) xenografts, a well-established model that enables real-time visualization of human cancer cell migration and reproduces conserved platelet–tumor interactions observed in humans. This study aimed to determine whether platelet-derived DEFA1/3 contributes to the aggressiveness of PDAC. Specifically, we hypothesized that platelets enriched in DEFA1/3 promote pro-tumorigenic and pro-migratory programs in PDAC cells. To test this hypothesis, we combined platelet transcriptomic re-analysis, in vitro functional assays, and in vivo zebrafish xenografts. Despite increasing evidence linking platelet-derived factors to tumor progression, the role of platelet defensins in pancreatic cancer biology remains largely unexplored. Here, we report that defensin-rich platelets (DRPs) promote proliferation, migration, and spheroid growth of pancreatic cancer cells, upregulate key aggressiveness-related genes, and are associated with adverse clinical outcomes, supporting the concept that platelet-derived DEFA1/3 represents a functional driver for PDAC progression and a candidate for translational exploration as a biomarker or therapeutic target... ´
- Line 56. Introduce in a greater detail the concept of tumor-educated platelets and expand the information on its role in PDAC.
Response: We thank the reviewer for this insightful comment. We have expanded the paragraph describing tumor-educated platelets to provide a more detailed explanation of their biological characteristics and specific role in PDAC. This revised section now includes examples such as ITGA2B and TGF-β to illustrate how TEPs contribute to epithelial–mesenchymal transition and tumor progression.
Introduction (page 2, lines 103-111)
´… The concept of tumor-educated platelets (TEPs) has emerged to describe platelets whose RNA and protein cargo are altered by tumor-derived signals. TEPs not only reflect the systemic influence of the tumor but also actively participate in shaping the tumor microenvironment through secreted mediators and intercellular communication(11,12). For instance, platelet-derived ITGA2B RNA has been identified as a robust biomarker for cancer detection, whereas platelet-secreted Transforming growth factor-beta (TGF-β) acts as a potent inducer of epithelial–mesenchymal transition (EMT), thereby directly promoting tumor progression(12,13).´
- Lines 56-60. The sentence is duplicated.
Response: We thank the reviewer for this observation. The duplicated sentence previously appearing between lines 56–60 has been removed to improve clarity and avoid redundancy. The paragraph has been revised to ensure smooth narrative flow and eliminate repetition.
- Line 63. EMT abbreviation should be introduced here when the epithelial-mesenchymal transition is used for the first time in the manuscript
Response: We thank the reviewer for this helpful observation. The abbreviation EMT (epithelial–mesenchymal transition) has now been introduced at its first mention in the manuscript (page 2, line 110) to ensure consistency with journal formatting guidelines.
- Line 73. Abbreviation Mks for megakaryocytes should be introduced above when the term is mentioned for the first time in the manuscript text. Double check the abbreviations in the entire manuscript. Some abbreviations are not provided in full (PTLS, etc.)
Response: We appreciate the reviewer’s careful observation. The abbreviation Mks (megakaryocytes) has now been introduced at its first mention in the manuscript (page 2, line 100). Additionally, we thoroughly reviewed the entire text to ensure that all abbreviations (e.g., PLTs, DRPs, PLPs) are defined upon first use and used consistently throughout the manuscript.
Materials and methods:
- 1. Protocols are not always sufficient enough to ensure reproducibility of this study. Expand them by providing detailed step-by-step instructions
Response: We thank the reviewer for this valuable suggestion. In response, we have thoroughly expanded the Materials and Methods section to ensure full reproducibility and transparency. Each protocol now includes detailed step-by-step descriptions specifying cell culture conditions, concentrations, incubation times, centrifugation parameters, and analytical software.
- 1. Cell culture (page 19, lines 1063-1084)
- The human 293T, AsPC-1, BxPC-3, Capan-2, and MEG-01 cell lines were obtained from the American Type Culture Collection (ATCC, Manassas, VA, USA). All cell lines were authenticated by the ATCC. Cells were cultured according to the ATCC recommendations. 293T cells were maintained in Dulbecco’s Modified Eagle Medium (DMEM; ATCC 30-2002) supplemented with 10% (v/v) FBS (Gibco, Cat. 16000044; Thermo Fisher Scientific, Waltham, MA, USA) and 1% penicillin–streptomycin (10,000 U/mL; Cat. No. 15140122. Thermo Fisher Scientific, Waltham, MA, USA). AsPC-1, BxPC-3, and MEG-01 cells were cultured in RPMI-1640 medium (ATCC 30-2001) containing 10% FBS and 1% penicillin–streptomycin. Capan-2 cells were cultured in McCoy’s 5A medium (ATCC 30-2007) supplemented with 10% FBS and 1% penicillin–streptomycin. All cultures were maintained at 37 °C in a humidified incubator with 5% CO₂ and passaged every 2–3 days to maintain exponential growth. For all experiments, cells were used between passages 5 and 20 to ensure consistency. Before treatment, cells were plated in T-75 flasks (Corning, NY, USA) or 6-well plates (Sigma-Aldrich, Saint Louis, MO, USA) and allowed to adhere overnight. The following day, the culture medium was replaced with a serum-free medium for 12 h to synchronize growth and minimize serum-derived variability. For defensin stimulation assays, cells were treated with recombinant human Defensin Alpha 1/3 (DEFA1/3; Abcam, Cambridge, UK, Cat. #ab97409) at a final concentration of 100 ng/mL for 72 h. Control (non-treated; Nt) cultures received the vehicle alone under identical conditions. All treatments were performed in biological triplicates, and each experiment was independently repeated at least three times to ensure reproducibility. All cell lines were mycoplasma-free.
- 2 Sample classification (page 19-20, lines 1085-1106)
To classify the platelet transcriptomic profiles of patients and controls, we imple-mented a Support Vector Machine (SVM) algorithm using the e1071 R package (v1.7-13) within R version 4.3.0 (R Foundation for Statistical Computing, Vienna, Austria). Nor-malized RNA-seq expression matrices were imported into R, and low-abundance genes (CPM < 1 in > 80% of samples) were filtered out to reduce noise. The remaining transcripts were log₂-transformed and scaled to zero mean and unit variance before model training. In the classification procedure, the SVM algorithm projected each sample into a high-dimensional feature space, where each axis represented a transcript. The position of a given sample along each axis corresponds to the normalized expression value of that gene. A One-Versus-One (OVO) strategy was applied to handle multiclass classification. In this approach, a separate binary SVM model is trained for each pair of classes, and the final class assignment for a given sample is determined by majority voting across all pairwise classifiers. The kernel function used was the radial basis function (RBF), which efficiently captures the nonlinear relationships between gene expression profiles. The cost (C) and gamma (γ) parameters were optimized using a 10-fold cross-validation within a grid search framework (tune.svm function). To evaluate the model performance, we used accuracy, F1-score, and area under the ROC curve (AUC), which were computed using the caret and pROC R packages. The misclassification error was assessed using confusion matrices generated from the withheld test samples (20% of the dataset, randomly selected). All analyses were conducted using reproducible R scripts, and the random seed was fixed at set.seed (1234) to ensure consistent classification results.
- 3 Differentiation of Mks (MEG-01) cells into platelet-like particles. (page 20, lines 1107-1126)
To obtain PLPs, MEG-01 megakaryoblast cells were differentiated in vitro following an optimized protocol adapted from Risitano et al., 2012(47) with minor modifications. Briefly, MEG-01 cells were seeded at a density of 2 × 10⁵ cells/mL in RPMI-1640 medium supplemented with 10% FBS and 1% penicillin/streptomycin. Recombinant human thrombopoietin (TPO) (PeproTech No. 300-18; Rocky Hill, NJ, USA) was added to the cultures at a final concentration of 100 ng/mL, and the cells were maintained at 37 °C in a humidified incubator with 5% CO₂. The differentiation process spanned 7 days, during which fresh TPO (100 ng/mL) was added every third day to sustain the megaryocytic maturation and proplatelet formation. Cell morphology was monitored daily using phase-contrast microscopy to confirm the appearance of cytoplasmic extensions and platelet-like protrusions. Following the differentiation period, the PLPs were purified using a stepwise centrifugation protocol. The cultures were centrifuged at 100 × g for 5 min to remove intact cells. The supernatant was carefully transferred and centrifuged again at 150 × g for 15 min to eliminate larger debris and fragments. Finally, PLPs were pelleted by centrifugation of the resulting supernatant at 750 × g for 15 min(18). The resulting pellet was resuspended in serum-free RPMI and quantified for protein concentration (Bradford assay) and particle counting using a Neubauer chamber under light microscopy. PLP morphology and purity were further verified by Giemsa staining prior to downstream assays.
- 4. Isolation of PTLS from healthy donor samples. (page 20, lines 1127-1147)
PLTs were isolated from voluntary laboratory donors under sterile conditions. Whole blood was collected into BD Vacutainer® sodium citrate tubes (Becton Dickinson, USA) to prevent coagulation, and samples were processed within 2 h of collection to ensure platelet integrity. Platelet-rich plasma (PRP) was obtained by centrifuging whole blood at 100 × g for 20 min at room temperature (RT) without a break. The upper PRP layer was carefully transferred to a new tube, and acetylsalicylic acid was added to a final concentration of 200 μM to inhibit the platelet activation. The PRP was centrifuged again at 100 × g for 20 min to eliminate residual red blood cells. Subsequently, the platelets were pelleted by cen-trifugation at 800 × g for 20 min at RT, and the supernatant was discarded. The resulting platelet pellet was gently resuspended in Tyrode’s buffer (pH 7.4) containing 0.1% bovine serum albumin (BSA) (Sigma-Aldrich; Cat. No. A2153). Platelet concentration and purity were assessed by manual counting using a Neubauer hemocytometer under light mi-croscopy, and morphology was verified by phase-contrast imaging(48). Purified PLTs were used immediately after isolation for all functional assays. PRP samples were obtained from voluntary laboratory donors who provided peripheral blood exclusively for in vitro research. No personal or clinical data were collected, and all samples were anonymized before processing. The use of human blood-derived material was conducted in accordance with institutional biosafety and ethical guidelines, under minimal-risk classification, and did not require formal informed consent according to national and international regulations (NOM-012-SSA3-2012; CIOMS, 2016).
- 5 Generation of DEFA1/3-rich platelet-derived particles (page 20-21, lines 1148-1170)
To generate DRPs, the megakaryoblastic cell line MEG-01 was genetically engineered using lentiviral transduction to overexpress DEFA1/3. Lentiviral particles were produced by co-transfecting 293T cells with Lipofectamine™ 2000 (Thermo Fisher Scientific, Cat. No. 11668019; Waltham, MA, USA) with the following plasmids: pLV-IRES-DEFA1/3 (donor vector containing the DEFA1/3 coding sequence inserted into the pLV-IRES backbone), pLJM1-GFP (transduction efficiency control), psPAX2 (Addgene, plasmid #12260; Watertown, MA, USA) (packaging plasmid), and pCMV-VSV-G (Addgene, plasmid #8454; Watertown, MA, USA) (envelope plasmid). Transfected cells were maintained in DMEM with 10% FBS for 48 h, and viral supernatants were collected every 12 h for 3 d, filtered through a 0.45 μm PES filter, and stored at 4 °C until use. MEG-01 cells were seeded at 2 × 10⁵ cells/mL and transduced with the viral supernatants at a multiplicity of infection (MOI) of 100, in the presence of 8 μg/mL polybrene to enhance infection efficiency. After 48 h, the medium was replaced, and the cells were cultured for an additional 72 h before puromycin (1 μg/mL) was applied for selection(49). Stable populations were obtained after three rounds of selection, yielding approximately 80% transduction efficiency as verified by GFP fluorescence. Following selection, MEG-01-DEFA1/3 and MEG-01-EV (empty vector control) cells were stimulated with recombinant TPO (200 ng/mL) for 72 h to induce the formation of PLPs. The released particles were collected by differential centrifugation, as described in Section 4.3, and designated as DRPs (DEFA1/3-rich platelets) and control platelets (PLPs-EV). Overexpression of DEFA1/3 in MEG-01-DEFA1/3 cells and DRPs was confirmed by qPCR and western blot analysis prior to downstream functional assays.
- 6 Genomic studies (page 21, lines 1172-1194)
Total RNA was extracted, and RT-PCR analysis was performed as previously described by Bandala et al. (2001)(50), with minor modifications to ensure reproducibility. Briefly, total RNA was extracted from AsPC-1, BxPC-3, MEG-01, PLPs-Ev, and DRPs cells using TRIzol™ reagent (Invitrogen). No. 15596026, USA) according to the manufacturer instructions. RNA concentration and purity were determined using a NanoDrop™2000c (Thermo Fisher Scientific, Waltham, MA, USA) spectrophotometer by measuring the A260/A280 ratio, and integrity was verified by 1% agarose gel electrophoresis. For reverse transcription, 1 μg of total RNA was retrotranscribed in a 20 μL final reaction volume using ThermoScript™ Reverse Transcriptase (Invitrogen) with random hexamer primers and reaction buffer supplied by the manufacturer (Invitrogen). Polymerase Chain Reaction (PCR) amplifications were performed using AmpliTaq Gold™ DNA polymerase (Applied Biosystems) in a 25 μL reaction volume containing 2.5 μL of 10× PCR buffer, 0.5 μL of dNTP mix (10 mM each), 1 μL of forward and reverse primers (10 μM each), and 1 μL of complementary DNA (cDNA) template. Thermal cycling was performed in a Veriti™ 96-Well Thermal Cycler (Applied Biosystems, Foster City, CA, USA) under the following conditions: initial denaturation at 95 °C for 10 min, followed by 30 cycles of 95 °C for 30 s, 50 °C for 45 s, and 72 °C for 1 min. Final extension: 72 °C for 5 min. PCR products were electrophoresed on 1–2% agarose gels stained with SYBR™ Safe DNA Gel Stain (Invitrogen, No. S33102) and visualized under UV light. Bands corresponding to the expected amplicon sizes were excised and purified using the QIAquick Gel Extraction Kit (Qiagen) for Sanger sequencing. Amplification efficiency and linearity were verified by plotting standard curves derived from serial dilutions of cDNA, confirming that the reactions proceeded within the logarithmic phase.
- 7 Transfection (page 21, lines 1195-1252)
For lentiviral particle production, 293T cells were co-transfected with the following plasmids: donor vector pLV-IRES-DEFA1/3, control vector pLJM1-GFP, packaging plasmid psPAX2, and envelope plasmid pCMV-VSV-G, following the manufacturer’s instructions for Lipofectamine™ 2000 (Invitrogen, USA). Before transfection, plasmid DNA was purified using an endotoxin-free Maxiprep kit (Qiagen, Cat. No. 12362. Germany) to ensure optimal transfection efficiency and to prevent cytotoxicity. For each 10 cm plate, 12 μg of total DNA (ratio 4:3:2:1 for pLV-IRES-DEFA1/3: psPAX2: pCMV-VSV-G: pLJM1-GFP) was mixed with Lipofectamine reagent in Opti-MEM™ medium and incubated for 20 min at room temperature before being added to the cells. After an 8-hour co-transfection period, the transfection medium was replaced with complete DMEM supplemented with 10% FBS, and the cells were incubated at 37 °C and 5% CO₂. Viral supernatants were collected at 48 and 72 h post-transfection, clarified by centrifugation at 500 × g for 10 min, and filtered through a 0.45 μm PES filter to remove cellular debris. The viral suspension was then concentrated by ultracentrifugation at 25,000 × g for 2 h at 4 °C using a SW32 Ti rotor (Beckman Coulter). The resulting viral pellet was resuspended in 1 mL of sterile Phosphate-Buffered Saline (PBS) containing 1% BSA and stored at −80 °C in single-use aliquots to avoid repeated freeze–thaw cycles. Viral titers were determined by transducing 293T cells with serial dilutions of the viral stock and quantifying GFP-positive cells after 72 h using fluorescence microscopy, as described by Schwarz-Cruz et al., 2020 (51). To determine the optimal transduction conditions, AsPC-1 cells were initially seeded to reach 80% confluence and infected with Lv-GFP lentiviral particles at different multiplicities of infection (MOI = 10, 20, 50, 100, and 200). After 48 h, GFP expression was evaluated using a Leica fluorescence microscope (Wetzlar, Germany). An MOI of 100 was selected as the optimal value for subsequent experiments, as it achieved robust transgene expression while minimizing cytotoxicity. Transduction was performed at 37 °C for 24 h, after which the virus-containing medium was replaced with fresh complete RPMI medium supplemented with 10% FBS. The cells were then incubated for an additional 48 h before initiating antibiotic selection. To generate stable cell lines, puromycin (2 μg/mL; Sigma-Aldrich, St. Louis, MO, USA) was added to the culture medium, which was refreshed every two days. Selection continued until all non-infected control cells were no longer viable (approximately 5–7 days). Surviving clones were expanded and maintained in 1 μg/mL puromycin to ensure stable transgene expression. The resulting cell populations were designated as MEG-01-DEFA1/3 and MEG-01-GFP, corresponding to cells stably expressing the DEFA1/3 transgene or GFP control, respectively(51).
- 8 Cell migration assays (pages 22-23, lines 1253-1270)
To evaluate the effects of DEFA1/3 and platelet-derived particles on the migratory capacity of pancreatic cancer cells, AsPC-1 and BxPC-3 cells were used. A total of 4 × 10⁵ cells were seeded into (Ibidi® 35 mm Culture Dishes containing Culture-Insert 2 Well systems (Cat. No. 80206; Gräfelfing, Germany) and allowed to form confluent monolayers within 24 h. Each insert well received 70 μL of cell suspension and was incubated at 37 °C in a humidified atmosphere containing 5% CO₂. To ensure that wound closure reflected cell migration rather than proliferation, the medium was supplemented with 5 μM cytosine β-D-arabinofuranoside hydrochloride (Ara-C; Sigma-Aldrich). Cat. No. C1768; St. Louis, MO, USA) as a DNA synthesis inhibitor. After the initial 24-hour attachment period, the Culture-Insert 2 Well was gently removed using sterile forceps to create a defined cell-free gap (wound). Cells were then treated under the following experimental conditions: Recombinant DEFA1/3 (100 ng/mL; Abcam, USA), PLP-Ev, DRPs, and Nt. The cultures were maintained for an additional 72 h under standard conditions. Wound closure was monitored by capturing five random fields per condition using an inverted phase-contrast microscope (Leica Microsystems, Wetzlar, Germany) at 0, 24, 48, and 72 h. Images were analyzed using the ImageJ software (NIH, USA). The percentage of wound closure was calculated as follows:
Each condition was tested in triplicate, and the results are presented as mean ± S.E.M from at least three independent experiments.
- 9 Spheroid culture. (page 23, lines 1274-1291)
Capan-2 and BxPC-3 pancreatic cancer cells were cultured using the liquid overlay technique, as previously described by Espinosa et al. (2012)(52), with minor modifications. Briefly, P60 Petri dishes were coated with a thin layer of 1% agarose (Sigma-Aldrich, Cat. No. A0169; St. Louis, MO, USA) (w/v) prepared in sterile PBS to prevent cell adhesion and allowed it to solidify at room temperature. Subsequently, 1 × 10⁶ cells, previously expanded as a monolayer, were seeded onto each agarose-coated plate in Leibovitz L-15 medium supplemented with 5% FBS (Gibco) and incubated at 37 °C in a humidified atmosphere. After 48 h, the floating cell aggregates began to form compact spheroids. Cultures were maintained for 7 days, during which the medium was gently refreshed every two days to preserve the nutrient balance and remove debris or aberrant (irregular) spheroids. To improve spheroid uniformity, the plates were transferred to a rotary shaker incubator (60 rpm) on the second day of culture, which promoted even shear forces and homogeneous spheroid formation. Spheroid growth was monitored daily under an inverted microscope (Leica Microsystems, Germany), and the diameters were quantified using a calibrated eyepiece graticule (Zeiss) or ImageJ software for digital images. At least 30 spheroids per condition were measured to determine the mean spheroid diameter ± S.E.M.
- 10 Viability assay. (page 23, lines 1292-1309)
Cell viability was evaluated using crystal violet (Sigma-Aldrich; Cat. No. C0775) staining, as previously described by Feoktistova et al. (2016)(53), with minor modifications. Briefly, 5 × 10³ cells per well were seeded in 200 μL of complete culture medium into 96-well flat-bottom plates (Corning, USA) and allowed to adhere overnight at 37 °C in 5% CO₂. At 0 and 72 h, the wells were gently washed twice with PBS to remove debris and non-adherent cells. Attached cells were fixed with 100% methanol for 10 min at room temperature and air-dried. Subsequently, the cells were stained with 0.05% (w/v) crystal violet solution (Sigma-Aldrich) for 15 min, followed by thorough washing with distilled water to remove excess dye and air-drying. To quantify cell viability, 33% (v/v) acetic acid was added to each well to solubilize the bound dye, and the optical density (OD) was measured at 570 nm using a microplate reader (Synergy HT; BioTek, Winooski, VT, USA). Relative cell viability (%) was calculated as follows:
All treatments were performed in triplicate wells across three independent experiments, and the results were expressed as mean ± S.E.M. Representative images of the stained wells were captured using a digital camera at 1× magnification before quantification.
4.11 Clonogenicity assay. (page 24, lines 1341-1353)
BxPC-3, AsPC-1, and Capan-2 cells were seeded at a low density (300–800 cells/well, depending on the cell line) in 6-well plates and allowed to adhere overnight. The cells were then treated with N), PLP-Ev, or DRPs and incubated for 10–14 days until visible colonies (>50 cells) were formed. At each passage, parallel wells were fixed with 4% paraformaldehyde (Sigma-Aldrich; Cat. No. 158127) for 10 min and stained with 0.05% crystal violet to confirm colony formation. For serial clonogenicity assessment, colonies were trypsinized, resuspended as single-cell suspensions, and re-plated at clonal density under identical conditions for two additional passages (P2–P3). At the final passage, the colonies were fixed and stained as described above. Colony numbers were quantified using ImageJ software (NIH, USA) and expressed as the survival fraction relative to the Nt. The persistence of colony-forming ability across passages is interpreted as evidence of self-renewal potential. All experiments were performed in triplicate across three independent assays.
- 12 Western Blot analysis. (page 24, lines 1355-1378)
Total protein lysates from PTLs, PLPs-Ev, and DRPs cells were extracted using RIPA lysis buffer (50 mM Tris-HCl, pH 7.4; 150 mM NaCl; 1% NP-40; 0.1% SDS; 0.5% sodium deoxycholate) supplemented with protease and phosphatase inhibitor cocktails (Sigma-Aldrich, Cat. #P8340 and #P0044, St. Louis, MO, USA). Protein concentrations were quantified using the Pierce™ BCA Protein Assay Kit (Thermo Fisher Scientific, Cat. #23225, Waltham, MA, USA) according to the manufacturer’s instructions. Equivalent amounts of total protein (20–30 µg) were mixed with 4× Laemmli sample buffer (Bio-Rad, Cat. #1610747) containing 5% β-mercaptoethanol and denatured at 95 °C for 5 min. Proteins were resolved by SDS-PAGE on 12% polyacrylamide gels and transferred onto 0.2 µm nitrocellulose membranes (Bio-Rad, Cat. #1620112) using a Trans-Blot® Turbo Transfer System (Bio-Rad Laboratories, Hercules, CA, USA). Membranes were blocked for 1 h at room temperature with Intercept® TBS Blocking Buffer (LI-COR, Cat. #927-60001) and incubated overnight at 4 °C with primary antibodies diluted in blocking buffer containing 0.1% Tween-20. The following primary antibodies were used: Anti-DEFA1/3 (mouse monoclonal, Abcam, Cat. #ab97409, 1:1000), anti-GAPDH (rabbit monoclonal, Cell Signaling Technology, Cat. #5174, 1:5000) was used as the loading control. After washing, membranes were incubated for 1 h at room temperature with species-specific infrared secondary antibodies: Goat anti-mouse IgG (IRDye® 680RD, LI-COR, Cat. #926-68070, 1:10 000), and goat anti-rabbit IgG (IRDye® 800CW, LI-COR #926-32211, 1:10 000). Fluorescent signals were detected using an Odyssey® CLx Infrared Imaging System (LI-COR Biosciences, Lincoln, NE, USA). Band intensities were quantified using Image Studio™ Lite software (LI-COR), and DEFA1/3 protein content was normalized to GAPDH. All Western blot assays were performed in biological triplicates to ensure reproducibility.
4.13 Zebrafish husbandry and ethical compliance. (pages 24-25, lines 1379-1407)
Adult zebrafish (Danio rerio) of the Tab-wik strain, originally provided by Dr. Ernesto Maldonado (Institute of Marine Sciences and Limnology, UNAM) and Dr. Francisco Carmona (Institute of Cellular Physiology, UNAM), were maintained under standard aquaculture conditions at 28.5 °C, with pH adjusted to 7.0–7.5, and a 14 h light/10 h dark cycle. Embryos were kept in embryo water (distilled water supplemented with 5 g/L Instant Ocean® Salt and 7.5 mg/L NaHCO₃).
General husbandry procedures for both adults and larvae followed established protocols detailed in The Zebrafish Book: A Guide for the Laboratory Use of Zebrafish (Danio rerio) (Westerfield, 2000)(54). Experimental procedures were conducted according to international animal care standards, including the Institutional Animal Care and Use Committee (IACUC) guidelines (University of Oregon), and were approved by the Internal Committee for the Care and Use of Laboratory Animals (CICUAL; Comité Interno para el Cuidado y Uso de Animales de Laboratorio, INSTITUTO NACIONAL DE MEDICINA GENOMICA (INMEGEN); approval no. 427). For euthanasia, embryos older than 4 days post-fertilization (dpf) were sacrificed by immersion in chilled water (2–4 °C) until opercular movements ceased, followed by additional exposure to the same chilled water for several minutes, as recommended by the American Veterinary Medical Association (AVMA) Guidelines for the Euthanasia of Animals (2020(55).
Animal care and monitoring. All procedures were designed to minimize the pain and distress. Embryos were anesthetized with 0.04% tricaine (MS-222; Sigma-Aldrich, Cat. No. E10521) during microinjection and was handled under a stereomicroscope to avoid mechanical stress. Embryos were monitored periodically during the 24 h post-injection period for signs of abnormal morphology, hemorrhage and death. Those presenting malformations or loss of viability were excluded from the analysis. No unexpected adverse events were observed. Humane endpoints were defined as death or severe malformation before imaging at 24 h post-injection, at which point the surviving embryos were imaged and subsequently maintained under standard conditions. Euthanasia was performed on larvae older than 4 dpf according to the AVMA guidelines.
- 14 Embryo preparation, microinjection, and migration assays. (pages 25-26, lines 1408-1463)
At 2 dpf, the embryos were manually dechorionated using Dumont No. 5 forceps and anesthetized in 0.04% tricaine (MS-222; Sigma-Aldrich, Cat. No. E10521). The embryos were positioned in 3% agarose-coated Petri dishes to ensure stability during injection. BxPC-3 pancreatic cancer cells pretreated with DRPs, PLPs, or left Nt were used for the xenotransplantation. Approximately 300 cells were injected into the yolk sac of each embryo. Each zebrafish embryo was considered an independent experimental unit. For each independent experiment, 20 embryos were allocated to each group (DRP, PLPs, and Nt), resulting in approximately 60 embryos per replicate and 300 embryos across all experiments. The chosen sample size followed previously validated zebrafish xenograft models (Martínez-López et al., 2021)(56,57) that demonstrated consistent tumor dissemination variability within this range, ensuring reproducibility and statistical reliability without requiring a formal a priori power calculation. Cells were pre-labeled with CellTracker™ Red CMTPX (Thermo Fisher Scientific, Cat. No. C34552), were resuspended at 1 x 103 cells/10 μL, and ~300 cells were injected into the yolk sac of each embryo, under blinded conditions, following xenotransplantation procedures adapted from Martínez-López et al. 2021(56,57). Borosilicate glass capillaries (10 cm length, 1.0 mm outer diameter (OD), 0.58 mm inner diameter (ID), without filament; Sutter Instrument) were pulled using a vertical pipette puller (Model 51210, Stoelting) and trimmed 0.8 cm from the shoulder using Dumont No. 5 forceps. Injections were performed using a Pneumatic Pico-Liter Injector (PLI-100A; Warner Instruments).
Post-injection, the embryos were rinsed with embryo water and incubated at 34 °C in Petri dishes containing sufficient embryo water, with a density of ≤30 embryos per dish. The inclusion criteria were defined a priori as (i) viable 2 dpf embryos and (ii) successful xenotransplantation verified immediately after injection (T0) by the presence of fluorescently labeled tumor cells at the yolk sac injection site. Embryos remained eligible for analysis regardless of whether fluorescent cells were confined to the yolk sac or had already disseminated at 24 h post-injection; migration was treated as an outcome. The exclusion criteria (predefined) were technical failure of injection (off-target delivery or widespread leakage at T0), severe malformation or hemorrhage, or death prior to imaging at 24 h. Each experimental group included exactly 20 embryos (n = 20 per group, per replicate); the exact n used in each analysis is reported in the corresponding figure legend. At 24 h post-injection, the embryos were examined in a blinded manner using a Zeiss epifluorescence microscope equipped with a Canon digital camera. Images were processed using FIJI–ImageJ software, and the number of fluorescent cells detected in the posterior trunk and tail (beyond the yolk sac) was quantified(56).
Randomization and control of confounders. Randomization was performed according to the treatment sequence (temporal alternation) during the injection process. Embryos from mixed clutches were selected without distinction and injected sequentially in alternating order across the treatment groups (DRP, PLP, and Nt) to avoid time-dependent or operator bias. Each experimental day included all three groups to minimize batch effects. All embryos were maintained under identical conditions (temperature, density ≤30 embryos per dish, and 14/10 h light/dark cycle). The plate positions within the incubator were rotated daily to prevent location bias. Outcome assessment was performed 24 h post-injection under blinded conditions. The operator was aware of the treatment allocation during the injection; however, outcome evaluation was performed using coded image files to reduce observer bias.
Outcome measures. The primary outcome measure was the number of fluorescent BxPC-3 cells that migrated beyond the yolk sac 24 h post-injection, quantified using FIJI–ImageJ. This variable directly assessed the pro-tumorigenic effects of DRPs in vivo. The secondary outcome measures included the fluorescent area corresponding to disseminated cells and the percentage of embryos showing successful xenografts. These measures were selected based on previous zebrafish xenotransplantation studies that evaluated tumor dissemination and invasion dynamics (Martínez-López et al., 2021)(32). The sample size (n = 20 embryos per group per replicate) was determined based on this primary outcome.
- 15 Transcriptomic analyses and survival modeling. (page 26, lines 1464-1475)
RNA-seq expression data and associated clinical annotations for PDAC were obtained from TCGA-PAAD via the Genomic Data Commons (GDC) Data Portal (https://portal.gdc.cancer.gov/). Raw counts were normalized to CPM and log₂-transformed using the limma/voom pipeline implemented in R (v4.3.0). For survival analyses, patients were stratified into quartiles based on DEFA1/3 expression, and OS was compared using Kaplan–Meier curves generated with the survival and survminer R packages. To evaluate potential interactions with circulating platelet levels, patients were further subdivided into four categories (high/low DEFA1/3 × high/low platelets) based on the median values of both variables. Log-rank tests were used to assess statistical significance. All analyses were conducted in R using publicly available TCGA data, following established computational reproducibility guidelines.
- 16 Single-sample gene set enrichment analysis. (pages 26-27, lines 1476-1519)
To explore functional associations, ssGSEA was performed using the Gene Set Variation Analysis (GSVA) R package (v3.21). Curated gene sets related to platelet activation, neutrophil infiltration, and EMT were used as reference pathways(58). Spearman’s rank correlation was applied to assess associations between DEFA1/3 expression and ssGSEA enrichment scores across samples. Correlation strength was quantified using the rho coefficient, and p-values were calculated to determine the significance. Scatter plots with linear regression lines were generated using the ggplot2 R package, with rho and p-values indicated in each panel(59). All analyses were conducted in R (v4.3.0) using publicly available TCGA-PDAC data.
- 17 Correlation analyses with aggressiveness-related genes. (pages 27, lines 1520-1535)
To further explore the molecular programs linked to PDAC aggressiveness, we selected a panel of genes previously reported to stratify tumor subtypes and progression (SPARC, GATA6, KDM6A, and EGR1). Transcriptomic scores for invasion (MMPs), motility (integrins), and EMT-like programs were computed using curated hallmark and Gene Ontology (GO) gene sets. Associations between gene expression and program scores were tested using Spearman’s correlation, and regression plots were visualized with confidence intervals(60).
To further explore the molecular programs associated with PDAC aggressiveness, we analyzed a curated panel of genes previously reported to stratify pancreatic tumor subtypes and progression (SPARC, GATA6, KDM6A, and EGR1) (61). Transcriptomic activity scores were calculated for invasion (MMP-related), motility (integrin-related), and EMT-like programs using hallmark and GO gene sets from the Molecular Signatures Database (MSigDB v7.5). Associations between gene expression and program scores were evaluated using Spearman’s rank correlation, and results were visualized as regression plots with 95% confidence intervals generated in R (v4.3.0).
- 18 Statistical analysis. (pages 27, lines 1537-1542)
Data are presented as mean ± S. E. M. M. from a minimum of three independent experiments, each conducted in triplicate. Group comparisons were performed using one-way analysis of variance (ANOVA), followed by Tukey’s post-hoc test to assess pairwise differences. Statistical significance was set at p < 0.01. All statistical analyses and graphical representations were performed using GraphPad Prism software version 9.0 (GraphPad Software, San Diego, CA, USA).
- Healthy donors used as a source of PTLS should be characterized (number, age, sex, etc.)
Response: We thank the reviewer for this comment. The PTLS used in this study were derived from voluntary laboratory donors who provided peripheral blood solely for in vitro experimentation. No personal identifiers or clinical data were collected, and all samples were anonymized. The use of these materials was conducted under minimal-risk classification following institutional and national ethical guidelines (NOM-012-SSA3-2012; CIOMS 2016).
(Methods 4.4, page 20, lines 1144-1147).
´…The use of human blood-derived material was conducted in accordance with institutional biosafety and ethical guidelines, under minimal-risk classification, and did not require formal informed consent according to national and international regulations (NOM-012-SSA3-2012; CIOMS, 2016). ´
- Any commercialized kits, reagents, instruments, software, antibodies, etc. used in the research, should be provided with their full name, along with the information of the manufacturers/suppliers/software details (Company Name, catalog number, City, Province/State, Country).
Response: We appreciate the reviewer’s suggestion. We have now included the full details (manufacturer, catalog number, city, and country) for all commercial kits, reagents, instruments, and software used throughout the study. These details have been incorporated in the (Methods 4.1-4.18 pages 19-27, lines 1061–1542).
- Equipment used is not always described
Response: We thank the reviewer for this observation. All instruments and software used in the study have now been explicitly identified in the Materials and Methods section, including manufacturer, model, and location. These details are provided within each corresponding subsection (Methods 4.1-4.18 pages 19-27, lines 1061–1542).
- Provide primer sequences used for PCR
Response: We appreciate the reviewer’s suggestion. The sequences of all primers used for PCR amplification have now been included in the revised version of the manuscript. Specifically, the primer sequences, target genes, amplicon sizes, and reference sources are provided in the newly added Supplementary Table S1 (“Primer Sequences Used for PCR”).
- No description of WB used
Response: We appreciate the reviewer’s observation. A detailed description of the Western blot procedure has now been incorporated in Section 4.12 (Western Blot analysis). The revised version specifies the buffer composition, antibody details (including catalog numbers, clones, sources, and dilutions), equipment used, and quantification methods.
Methods (Section 4.12, page 24, lines 1355-1378).
Results:
- Scale bars are not shown
Response: We thank the reviewer for this valuable observation. Scale bars have now been added to all microscopy and imaging panels in the revised figures (Figures 2–6, supp fig 1-2 ). Each scale bar corresponds to the magnification used for image acquisition and was calculated according to the specifications of the Leica microscope software used during data collection. The inclusion of scale bars improves figure accuracy and allows for better comparison of spatial dimensions across experimental conditions.
- WB images should be presented uncut. Provide original WB images
Response: We thank the reviewer for this helpful suggestion. In response, the original, uncut Western blot images have now been included in the Supplementary Material (SM1) to ensure full transparency and traceability of the data. The main figures now display cropped bands exclusively for clarity, while the complete gels are available for reference.
- 3. Figure captions should contain the list of abbreviations
Response: We thank the reviewer for this valuable observation. In response, we have revised all figure captions to include the list of abbreviations corresponding to the terms and symbols used in each figure. This modification ensures clarity and consistency throughout the manuscript, in line with the journal’s formatting requirements.
(Figure legend 1, pages 4-5, lines 299-325).
(Figure legend 2, page 7, lines 413-440).
(Figure legend 3, pages 8-9, lines 546-572).
(Figure legend 4, pages 10-11, lines 613-641).
(Figure legend 5, pages 12-13, lines 724-751).
(Figure legend 6, pages 14-15, lines 780-823).
(Figure legend 7, pages 15-16, lines 845-867).
(Supplementary Figure 1, anexo1).
(Supplementary Figure 2, anexo1).
(Supplementary Figure 3, anexo1).
- Figure 3. Figure caption should be significantly expanded to represent “independent” sources of the information. They should contain the description of what and how was measured, key findings, statistical analysis used, the way numerical data are presented (e.g., mean and SD, number of replicates, etc. Quality of images is insufficient. Scale bar is not visible. In general, captions are often not sufficient to understand what is shown.
Response: We sincerely thank the reviewer for this detailed and constructive comment. The caption of Figure 3 has been substantially expanded to provide a full and independent description of the experimental setup, including what and how measurements were performed, the number of replicates, and the statistical analysis applied (one-way ANOVA with Tukey’s post-hoc test). Quantitative data are now presented as mean ± S.E.M., with the number of independent experiments clearly stated. Furthermore, a scale bar has been incorporated into the representative microscopy images, and visual quality has been improved for clarity and reproducibility. In addition, these same improvements and formatting standards were systematically applied to all other figure legends to ensure consistency, completeness, and compliance with the journal’s requirements for independent interpretability of figures. (Figure 3 pages 8-9, lines 546-572).
- Precise p values should be mentioned in Figures
Response: We thank the reviewer for this valuable suggestion. We have revised all figure captions to include the exact p-value thresholds and standardized the notation across all figures. Statistical significance is now reported as follows: *p ≤ 0.05, **p ≤ 0.01, and ***p ≤ 0.001. These updates were consistently applied throughout the manuscript to improve clarity and ensure transparency in the presentation of statistical results.
(Figure legend 1, pages 4, line 302 and 323).
(Figure legend 2, page 7, line 433).
(Figure legend 3, page 9, line 566).
(Figure legend 4, pages 10, lines 616).
(Figure legend 5, pages 12-13, lines 736-743).
(Figure legend 6, pages 15, line 817).
(Figure legend 7, pages 16, lines 851).
(Supplementary Figure 1, anexo1).
(Supplementary Figure 2, anexo1).
(Supplementary Figure 3, anexo1).

Round 2
Reviewer 1 Report
Comments and Suggestions for Authors
The authors have addressed all of my comments.
Reviewer 2 Report
Comments and Suggestions for Authors
The authors have addressed the comments
Comments on the Quality of English Language- English level should be improved. The manuscript contains multiple grammatical and stylistic errors, typos, etc.